# A versatile pipeline for the multi-scale digital reconstruction and quantitative analysis of 3D tissue architecture

**Hernán Morales-Navarrete[1], Fabián Segovia-Miranda[1], Piotr Klukowski[1], Kirstin Meyer[1], Hidenori Nonaka[1,2], Giovanni Marsico[1], Mikhail Chernykh[1], Alexander Kalaidzidis[1], Marino Zerial[1]\*, Yannis Kalaidzidis[1,3]\***

[1]Max Planck Institute of Molecular Cell Biology and Genetics, Dresden, Germany; [2]Rohto Pharmaceutical, Tokyo, Japan; [3]Faculty of Bioengineering and Bioinformatics, Moscow State University, Moscow, Russia

**Abstract** A prerequisite for the systems biology analysis of tissues is an accurate digital three-dimensional reconstruction of tissue structure based on images of markers covering multiple scales. Here, we designed a flexible pipeline for the multi-scale reconstruction and quantitative morphological analysis of tissue architecture from microscopy images. Our pipeline includes newly developed algorithms that address specific challenges of thick dense tissue reconstruction. Our implementation allows for a flexible workflow, scalable to high-throughput analysis and applicable to various mammalian tissues. We applied it to the analysis of liver tissue and extracted quantitative parameters of sinusoids, bile canaliculi and cell shapes, recognizing different liver cell types with high accuracy. Using our platform, we uncovered an unexpected zonation pattern of hepatocytes with different size, nuclei and DNA content, thus revealing new features of liver tissue organization. The pipeline also proved effective to analyse lung and kidney tissue, demonstrating its generality and robustness.

**\*For correspondence:** zerial@mpi-cbg.de (MZ); kalaidzi@mpi-cbg.de (YK)

**Competing interests:** The authors declare that no competing interests exist.

## Introduction

A major challenge for the understanding of mammalian tissue structure and function is the ability to monitor cellular processes across different levels of complexity, from the subcellular to the tissue scale (*Megason and Fraser, 2007*). This information can then be used to develop quantitative functional models that describe and predict the system behaviour under perturbed conditions (*Hunter et al., 2008*; *Smith et al., 2011*; *Fonseca et al., 2011*; *Sbalzarini, 2013*). The development of such multi-scale models requires first a geometrical model of the tissue, that is, an accurate three-dimensional (3D) digital representation of the cells in the tissue as well as their critical subcellular components (*Peng et al., 2010*; *Boehm et al., 2010*; *Mayer et al., 2012*). This can be constructed from high-resolution microscopy images with multiple fluorescent markers, either fusion proteins or components detected by antibody staining. Since organelles can be as small as ~0.1 μm in size, the geometrical model has also to cover a wide range of scales spanning over three orders of magnitude. However, substantial limitations persist with respect to availability of markers, volume of tissue to reconstruct, scale of measurements, computational methods to perform the analysis and sample throughput. Although a few existing platforms provide standard tools for 3D segmentation and methods to process 2D surface layers of cells [ImageJ/Fiji (*Girish and Vijayalakshmi, 2004*; *Collins, 2007*), ICY (*de Chaumont et al., 2012*) and MorphoGraphX (*Barbier de Reuille et al., 2015*)], the challenges posed by dense and thick tissue specimens require the development of new algorithms. Therefore, there is a demand for a platform that can provide the required set of methods for

**eLife digest** Understanding how individual cells interact to form tissues in animals and plants is a key problem in cell and developmental biology. To be able to answer this question researchers need to use microscopy to observe the cells in a tissue, extract structural information from the images, and then generate three-dimensional digital models of the tissue. However, the software solutions that are currently available are limited, and reconstructing three-dimensional tissue from microscopy images remains problematic.

To meet this challenge, Morales-Navarrete et al. extended the free software platform called MotionTracking, which had been used previously for two-dimensional work. The software now combines a series of new and established algorithms for analysing fluorescence microscopy images that make it possible to identify the different structures that make up a tissue and then create and analyse a three-dimensional model.

Morales-Navarrete et al. used the software to analyse liver tissue from mice. The resulting model revealed that liver cells called hepatocytes are arranged in particular zones within the tissue according to their size and DNA content. The software was also applied successfully to analyse lung and kidney tissue, which demonstrates that the approach can be used to create three-dimensional models of a variety of tissues.

Morales-Navarrete et al.'s approach can rapidly generate accurate models of larger tissues than were previously possible. Therefore, it provides researchers with a powerful tool to analyse the different features of tissues. This tool will be useful for many areas of research: from understanding of how cells form tissues, to diagnosing diseases based on the changes to features in particular tissues.

the reconstruction of multi-scale digital 3D geometrical models of mammalian tissues from confocal microscopy images.

The number of fluorescent markers that can be used simultaneously is limited to 4–5, making the reconstruction of tissue models a challenging problem. For a meaningful model, it is necessary to properly identify the different cell types within the tissue but also to detect subcellular and extracellular structures, for example, nuclei, plasma membrane or cell cortex, extracellular matrix (ECM) and cell polarity. Automated morphological cell recognition is a possible way to reconstruct dense tissue with limited number of markers.

Geometrical digital models of tissues also require 3D information over large volumes. Validated fluorescent protein chimeras are not always available, especially in the appropriate combination of fluorescence emission spectral profiles. On the other hand, in dense tissues immunostaining is inhomogeneous due to restricted antibody penetration. The development of protocols that render tissues optically transparent and permeable to macromolecules without significantly compromising their general structure enables the imaging of relatively thick specimens (*Chung and Deisseroth, 2013*; *Ke et al., 2013*). However, in the case of a densely packed tissue, for example, liver, homogeneous staining is still limited to a thickness of ~100 μm. Therefore, obtaining high-resolution data from large volumes of tissue (typically from 0.1 mm to a few centimetres) requires sectioning the sample into serial 100-μm-thick slices that are stained and imaged separately. Furthermore, the cutting process introduces artefacts, such as bending, uneven section surfaces and partial damage of tissue, that require corrections during tissue model reconstruction. Unfortunately, the publicly available generic image processing software is unable to deal with such problems.

In this study, we addressed these challenges by developing a set of new algorithms as well as implementing established ones in an adjustable pipeline implemented in stand-alone freely available software (http://motiontracking.mpi-cbg.de). As proof of principle, we tested the pipeline on the reconstruction of a geometrical model of liver tissue. We chose this particular tissue due to its utmost importance for basic research, medicine and pharmacology. In order to test the accuracy of the pipeline, we created a benchmark for the evaluation of dense tissue reconstruction algorithms comprising a set of realistic 3D images generated from the digital model of liver tissue. Furthermore,

we applied the platform to the analysis of lung and kidney tissue, demonstrating its generality and robustness.

## Results

Despite its importance and a long history of histological studies, only a few geometrical models of liver tissue have been published (*Hardman et al., 2007*; *Hoehme et al., 2010*; *Hammad et al., 2014*). The liver is composed of functional units, the lobules. In each lobule, bile canaliculi and sinusoidal endothelial cells form two 3D networks between the portal vein (PV) and the central vein (CV). The bile canalicular (BC) network is formed by hepatocytes and transports the bile, whereas the sinusoidal endothelial network transports the blood. The liver tissue has a number of remarkable features. One is the zonation of metabolic functions due to the fact that the hepatocytes located in the vicinity of the PV do not have the same metabolic activities as the hepatocytes located near the CV (*Kuntz and Kuntz, 2006*). Second, hepatocytes are remarkably heterogeneous in terms of number of nuclei (mono- and bi-nucleated) and ploidy (*Martin et al., 2002*; *Guidotti et al., 2003*; *Faggioli et al., 2011*). Third, the lobules contain two additional important cell types, stellate and Kupffer cells (*Baratta et al., 2009*).

To analyse the 3D organization of liver tissue, we established a workflow for confocal imaging of mouse liver specimens and developed an adjustable pipeline of new and established image analysis algorithms to process the images and build digital models of the tissue (*Figure 1* and *Figure 1—figure supplement 1*). First, we established a protocol for the preparation of tissue specimens for single- and double-photon confocal microscopy at different resolutions. To cover multiple scales from subcellular organelles to tissue spanning over three orders of magnitude, we used a 3D multi-resolution tissue image acquisition approach (*Figure 1A*). This consisted of imaging a tissue sample at low resolution (1 µm × 1 µm × 1 µm per voxel) and zooming on the parts of interest at high resolution (0.3 µm × 0.3 µm × 0.3 µm per voxel). Second, the multi-scale reconstruction of tissue architecture was obtained following the pipeline of *Figure 1B* and *Figure 1—figure supplement 1*. Briefly, (1) images were filtered using a novel Bayesian de-noising algorithm; (2) individual low-resolution images of each physical section were assembled in 3D mosaics; (3) tissue deformations caused by sample preparation were corrected; (4) large vessels were segmented; (5) the 3D mosaics of sections were combined in a full-scale low-resolution model; (6) high-resolution images were registered into the low-resolution one; (7) sinusoidal and BC networks as well as nuclei were segmented and, finally, (8) the different cell types were identified, classified and segmented. We used the geometrical model to provide a detailed and accurate quantitative description of liver tissue geometry, including the complexity of the sinusoidal and BC networks, hepatocyte size distribution, stellate and Kupffer cells distribution in the tissue. Additionally, our platform comprises a set of methods for the proper statistical analysis of different morphometric parameters of the tissue as well as their spatial variability (*Figure 1C*).

### Sample preparation and multi-resolution tissue imaging

Mouse livers were fixed by trans-cardial perfusion instead of the conventional immersion fixation (*Burton et al., 1987*) to minimize the time lag between the termination of blood flow and fixation (*Gage et al., 2012*). This proved to be absolutely essential to preserve the tissue architecture and the epitopes for immunostaining. Serial sections of fixed tissues were prepared at a thickness of 100 µm to maximize antibody penetration and limit laser light scattering. Liver sections were stained to visualize key subcellular and tissue structures, namely nuclei (DAPI), the apical surfaces of hepatocytes (CD13), the sinusoidal endothelial cells (Flk1) or ECM (Laminin and Fibronectin) and the cell cortex (F-actin stained by phalloidin). We tested various reagents and protocols to clear the liver tissue, such as glycerol and 2,2'thiodiethanol (TDE), and found that SeeDB (*Ke et al., 2013*) yielded the best results. Stained sections were imaged sequentially (generating Z-stacks) by one- and two-photon laser scanning confocal microscopy to maximize the number of fluorescent channels available. The same section was imaged twice, at low and high magnification, using 25×/0.8 and 63×/1.3 objectives, respectively. The first covers a large volume to reconstruct the whole lobule and the latter focuses on a small area to reconstruct the tissue at high resolution. The registration of 3D high-resolution images within low-resolution ones provides tissue-scale context information that is essential for the interpretation of the data at the cellular and subcellular level.

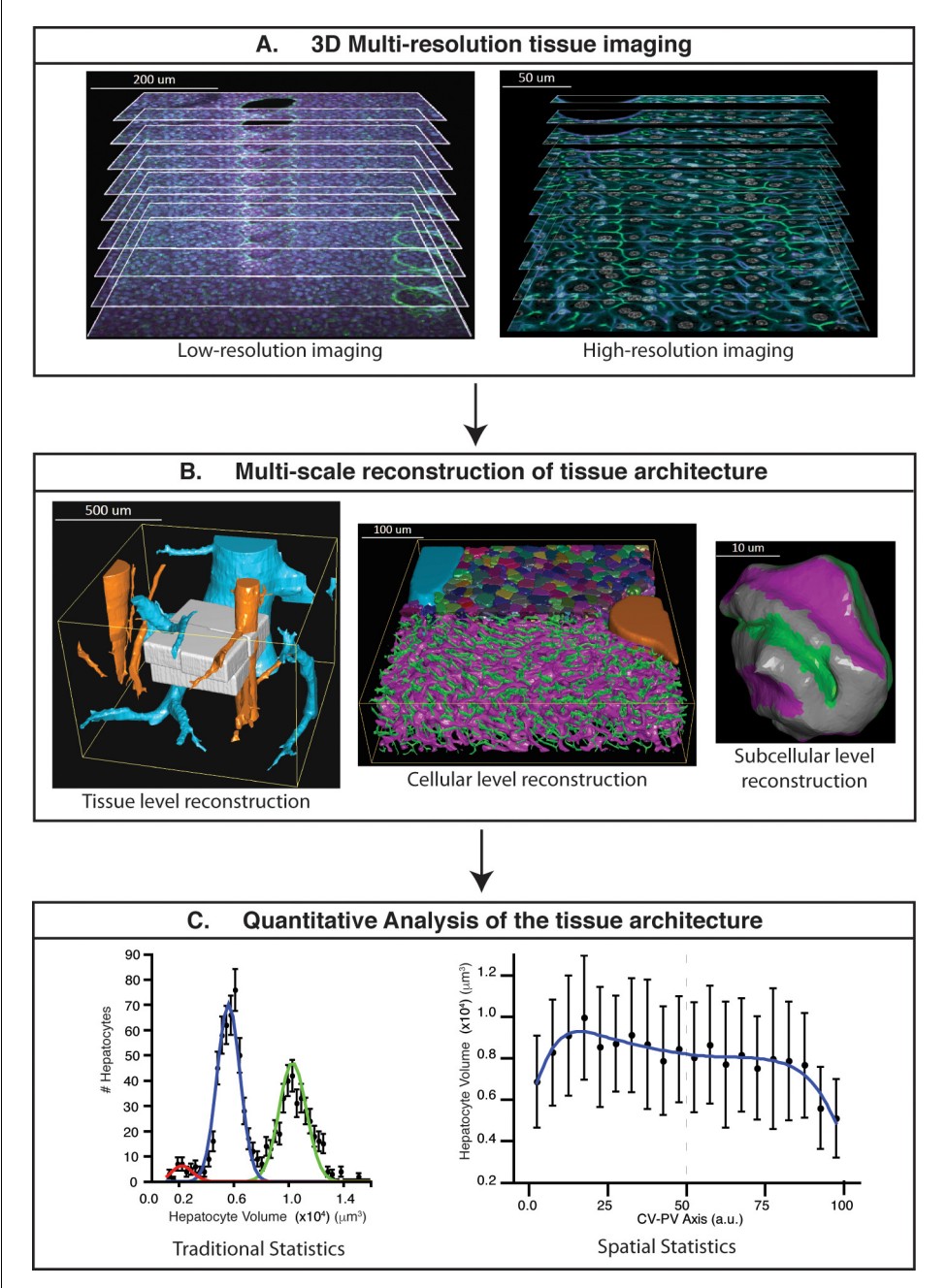

**Figure 1.** Schematic representation of the proposed pipeline. (**A**) 3D multi-resolution image acquisition: example of arrays of 2D images of liver tissue acquired at different resolutions. Low- (1 µm × 1 µm × 1 µm per voxel) and high- (0.3 µm × 0.3 µm × 0.3 µm per voxel) resolution images on the left and right sides, respectively. (**B**) Multi-scale reconstruction of tissue architecture: on the left, reconstruction of a liver lobule showing tissue-level information, i.e., the localization and relative orientation of key structures such as the portal vein (PV) (orange) and central vein (CV) (light blue). The high-resolution images registered into the low-resolution one are shown in white. On the middle, a cellular-level reconstruction of liver showing the main components forming the tissue, i.e., bile canalicular (BC) network (green), sinusoidal network (magenta) and cells (random colours). The reconstruction corresponds to one of the high-resolution cubes (white) registered on the liver lobule reconstruction (left side). On the right, reconstruction of a single hepatocyte showing subcellular-level information, i.e., apical (green), basal (magenta) and lateral (grey) contacts. (**C**) Quantitative analysis of the tissue architecture: example of the statistical analysis performed over a morphometric tissue parameter (hepatocyte volume) using the information extracted from the multi-scale reconstruction. On the left, hepatocyte volume distribution over the sample (traditional statistics). On the right, spatial variability (spatial statistics) of the same parameter within the liver lobule. Our workflow allows not only to perform traditional statistical analysis of different morphometric parameters but also to perform spatial characterizations of them. The graphs were generated from the

*Figure 1 continued on next page*

*Figure 1 continued*

analysis of one high-resolution cube of the multi-scale reconstruction (the one shown in middle of panel B). Boundary cells were excluded from the analysis.

The following figure supplements are available for figure 1:

**Figure supplement 1.** Workflow for the multi-scale reconstruction of tissue architecture from multi-resolution confocal microscopy images.

**Figure supplement 2.** Probabilistic image de-noising algorithm for 3D images.

**Figure supplement 3.** Optimal parameter selection.

**Figure supplement 4.** Comparison of our 3D image de-noising algorithm (BFBD) with standard methods in the field.

**Figure supplement 5.** Comparison of our 3D image de-noising algorithm (BFBD) with 'pure denoise' (PD) (*Luisier et al., 2010*) and 'edge preserving de-noising and smoothing' (EPDS) (*Beck and Teboulle, 2009*).

## Bayesian foreground/background discrimination (BFBD) de-noising

A major problem for the image analysis of thick tissue sections is the low signal-to-noise ratio deep into the tissue, especially for stainings that yield high and diffuse background (e.g. actin staining with phalloidin throughout the cytoplasm). To address this problem, we developed a new Bayesian de-noising algorithm that first makes a probabilistic estimation of the background and separates it from the foreground (see 'Methods'). Subsequently, the estimated background and foreground signals are independently smoothed and summed to generate a new de-noised image (*Figure 1—figure supplement 2*). We applied BFBD de-noising to both low- and high-resolution images. BFBD de-noising provides better results than the standard ones in the field, such as median filtering, Gauss low-pass filtering and anisotropic diffusion (*Figure 1—figure supplement 4*), but also outperforms (by quality and computational performance) other algorithms, known to be more elaborate, such as the 'Pure Denoise' (*Luisier et al., 2010*) and 'edge preserving de-noising and smoothing' (*Beck and Teboulle, 2009*) (see 'Methods') (*Figure 1—figure supplement 5*).

## Reconstruction of multi-scale tissue images

The tissue was imaged at low- and high-resolution for the multi-scale reconstruction. The reconstruction was performed in three steps: (1) images of physical sections were assembled as mosaics of low-resolution images, (2) all mosaics were corrected for physical distortions and combined in a single 3D image (image stitching) and (3) the high-resolution images were registered into the low-resolution one.

In more detail, the partially overlapping (~10% overlap) low-resolution images of each physical section were combined in 3D mosaics (*Figure 2A* and *Figure 2—figure supplement 1A*) using the normalized cross-correlation (NCC) approach (see 'Methods'). NCC was chosen because it allows finding accurate shifts given a coarse initial match between 3D images (*Emmenlauer et al., 2009*; *Peng et al., 2010*; *Bria and Iannello, 2012*). Then, the 3D image mosaics were combined into a single 3D image. The mechanical distortion and tissue damage produced by sectioning are such (as illustrated in *Figure 2B* and *Figure 2—figure supplement 1C*) that even advanced and well-established methods for image stitching (*Preibisch et al., 2009*; *Saalfeld et al., 2012*; *Hayworth et al., 2015*) fail due to the lack of texture correlations between adjacent sections. To address this problem, we developed a Bayesian algorithm for stitching images of bended and partially damaged soft tissue sections. The algorithm first corrects section bending and then uses the empty space at the interior of large structures (e.g. vessels) within adjacent sections to register and stitch them.

A prerequisite for the correction of section bending is the detection of its upper and lower surfaces (*Figure 2B*). The high degree of image axial blurring in thick samples (*Nasse and Woehl, 2010*) and the presence of large vessels pose problems for the detection of surfaces (see *Figure 2—figure supplement 1C*). The algorithm reconstructed the probability distribution of the surface excursion (deviation from the mean position over the neighbourhood) and then used it to predict the localization of each point at the surface (see 'Methods'). The surface predicted by the algorithm closely

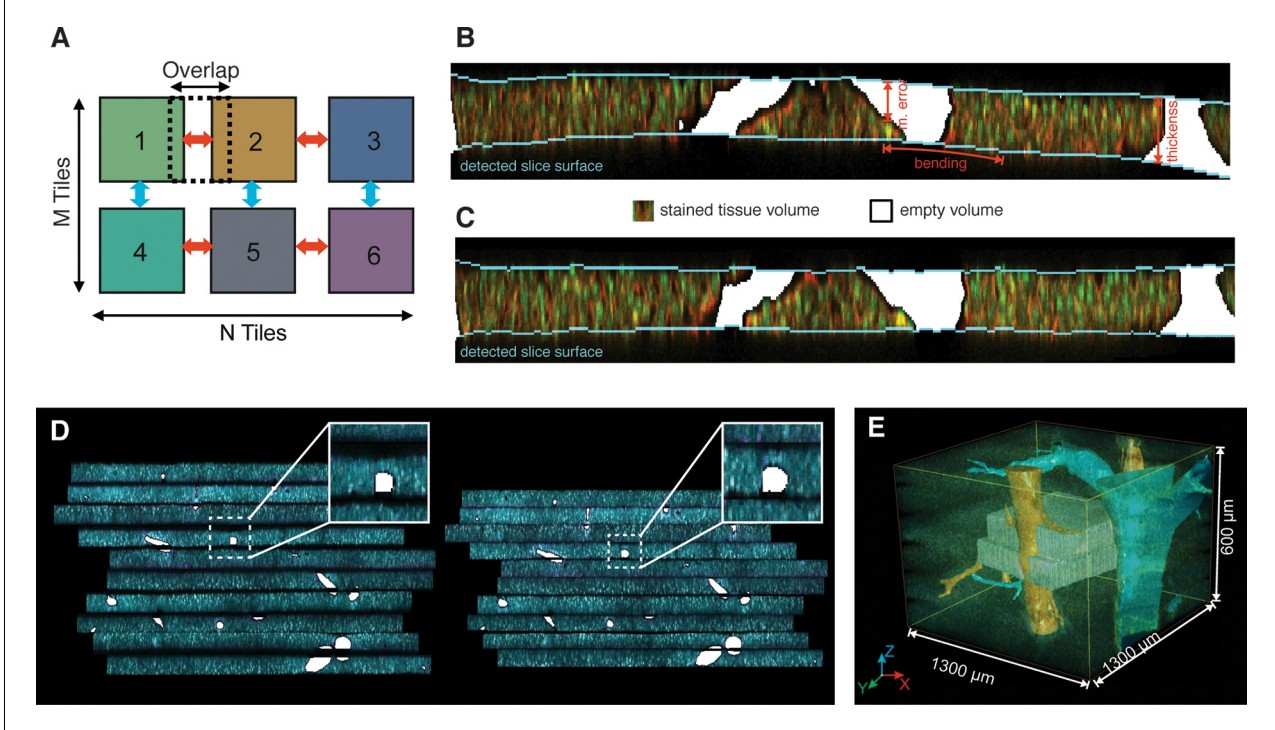

**Figure 2.** Reconstruction of a multi-scale lobule image. (**A**) Schematic representing a single serial section obtained from a grid of M × N partially overlapping 3D images (tiles). The cross-correlation between two neighbouring tiles in the grid provides a local metric, which describes the value of their relative shifts. The reconstruction of each section was performed by maximizing the sum correlations of each tile to all adjacent tiles (see 'Methods' for details). (**B, C**) Correction of tissue deformations (introduced during the sample preparation process) using a surface detection algorithm and β-spline transformation. (**B**) Output of the surface detection algorithm. The proposed Bayesian approach uses prior information about expected bending of the section, its thickness and measurement error (see 'Methods' for details) to determine the volume of the image belonging to the tissue and to the out-of-field region. (**C**) The tissue section after correcting its bending by using quadratic β-splines. (**D**) Tissue section before (left) and after (right) the correction of the mechanical distortions and the tissue damage. (**E**) Full lobule-level reconstruction established by the alignment of six low-resolution sections (1 µm × 1 µm × 1 µm per voxel) and the interpolation of blood vessels. Two high-resolution images (0.3 µm × 0.3 µm × 0.3 µm per voxel) were registered in the low-resolution reconstruction and are shown in grey (see *Video 1*).

The following figure supplements are available for figure 2:

**Figure supplement 1.** Reconstruction of multi-scale tissue images.

**Figure supplement 2.** Reconstruction of multi-scale tissue images.

matched the surface detected manually (*Figure 2—figure supplement 1G*). Then, the bending correction was performed by standard β-spline transformation (*Figure 2C,D*).

Next, the individual sections were combined. Since approximately one cell layer is removed upon sectioning, direct matching of two adjacent sections is impossible. Therefore, we first segmented the large vessels and then aligned the sections by matching them (*Figure 2D*). The vessels were segmented by using the local maximum entropy (LME) approach (*Brink, 1996*) (see 'Methods'). Subsequently, the segmented vessels were classified (marked as PV or CV) revealing the precise arrangement of lobule-level structures. Finally, we interpolated these vessels within the gaps caused by tissue removal by tri-linear intensity approximation.

Following the assembly of the low-resolution model, we registered the high-resolution images within it using rigid body transformation. To accelerate the search for registration parameters, we built a hierarchy of binned images and performed registration sequentially from the coarsest to the finest one (see 'Methods'). This method was used for the reconstruction of a liver tissue model from six serial sections, each imaged as a 3 × 3 mosaic grid with 10% overlap and resolution of 1 µm × 1 µm × 1 µm per voxel. Then, two sections, each imaged as a 2 × 2 mosaic grid at high-resolution

(0.3 μm × 0.3 μm × 0.3 μm per voxel) were registered within the low-resolution model. The reconstruction covers about 1300 μm × 1300 μm × 600 μm of the tissue and is presented in *Figure 2E* and *Video 1*.

## 3D image segmentation and active mesh tuning for the accurate reconstruction of tubular networks (sinusoids and BC) and nuclei

The next step was to reconstruct the tubular structures present in the tissue, that is, sinusoidal and BC networks. One of the most popular tools for image segmentation is global thresholding (*Pal and Pal, 1993*). In particular, the maximum entropy approach has been widely applied to image reconstruction problems, including the segmentation of fluorescent microscopy images (*Dima et al., 2011*; *Pecot et al., 2012*). However, since 3D confocal images are usually heterogeneous in intensity due to staining unevenness and light scattering in the tissue (*Lee and Bajcsy, 2006*), global thresholding approaches may produce segmentation artefacts. In contrast, local thresholding allows adjusting the segmentation threshold to the spatial variability. We applied the LME method to find segmentation thresholds in the de-noised images. For this, we split the 3D image into a set of cubes and calculated the maximum entropy segmentation threshold (*Brink, 1996*) within each cube. The threshold values were tri-linearly interpolated to the entire 3D image.

However, this segmentation approach produced two major artefacts. The objects were moderately swollen and contained holes resulting from local uneven staining. We used standard approaches to close the holes by morphological operations (opening/closing), which unfortunately led to even higher overestimation of the diameter of thin structures, such as sinusoids and BC. To correct this, we extended the segmentation algorithm by including the following steps. We generated a triangulation mesh of the segmented surfaces by the cube marching algorithm (*Lorensen and Cline, 1987*) (*Figure 3A*). Then, we tuned the active mesh so that the triangle mesh vertexes aligned to the maximum gradient of fluorescence intensity in the original image (*Figure 3A*). Finally, we generated a representation of the skeletonized image via a 3D graph describing the geometrical and topological features of the BC and sinusoidal networks. The reconstruction of sinusoidal and BC networks are shown in *Figure 3B,C*, respectively.

Nuclei were reconstructed similar to the tubular structures. However, as shown in *Figure 3—figure supplement 1A,B*, closely packed nuclei are optically not well-resolved in 3D confocal images, resulting in artificially merged structures. Since 30–60% (depending on the animal strain and age) of hepatocytes in adult liver are bi-nucleated, artificial nuclei merging compromises the tissue analysis. To address this problem, we used a probabilistic algorithm for double- and multi-nuclei splitting (*Figure 3—figure supplement 1*). Briefly, the algorithm first discriminated between mono-, double and multi-nuclear structures by learning the misfit distribution of triangulation mesh and nuclei approximation by single and double ellipsoids (*Figure 3—figure supplement 1A–G*). Then, the seed points for the multi-nuclear structures were detected using the Laplacian-of-Gaussian (LoG) scale-space maximum intensity projection (*Stegmaier et al., 2014*) and, finally, the real nuclear shapes were found using an active mesh expansion starting from the nuclei seeds (see 'Methods' for details). Tested in both synthetic and real 3D images, the algorithm proved capable of splitting clustered nuclei with different degrees of overlap (*Figure 3—figure supplement 1K*) with an accuracy of over 90%. Although this approach is based on active triangulation mesh, it achieved similar accuracy values to other recently published voxel-based methods for nuclei segmentation (*Amat et al., 2014*; *Chittajallu et al., 2015*).

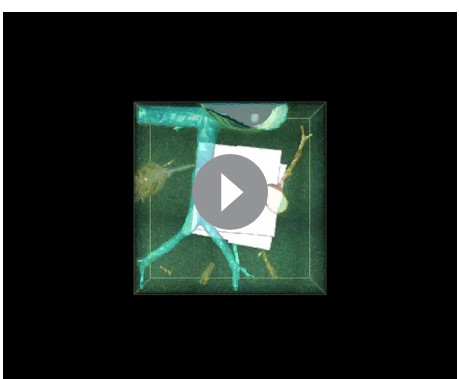

**Video 1.** 3D image visualization of a multi-resolution geometrical model of liver tissue. A set of six low-resolution (1.0 μm × 1.0 μm × 2.0 μm per voxel) and two high-resolution tissue sections (0.3 μm × 0.3 μm × 0.3 μm per voxel) were used. Central veins are shown in light blue, portal veins in orange and high-resolution cubes in grey.

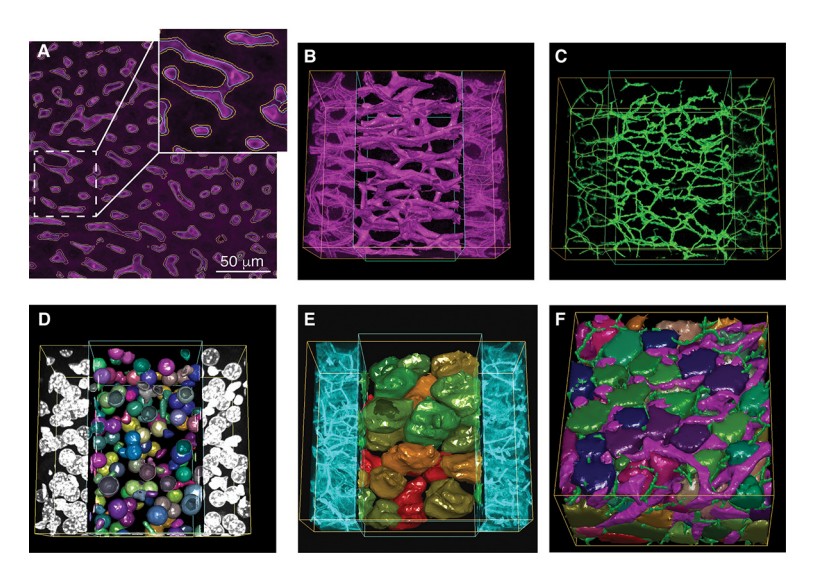

**Figure 3.** Reconstruction of tubular structures, nuclei and cells. (A) A single 2D image section is shown with the contours of the sinusoidal network reconstruction overlaid on the de-noised image. The contours of the initial mesh are drawn in yellow, and the ones of the tuned mesh are drawn in cyan. (B–E) 3D representation of the different structures segmented in a sample of liver tissue: sinusoids (B), BC (C), nuclei (D) and cells (E). All the reconstructed structures are shown together in (F). The reconstructed triangle meshes are drawn inside the inner box and the raw images are outside. In the case of tubular networks (i.e. sinusoids and BC), the central lines of the structures are shown together with the raw images.

The following figure supplements are available for figure 3:

**Figure supplement 1.** Nuclei splitting.

**Figure supplement 2.** Cell classification.

**Figure supplement 3.** Cell classification accuracy.

**Figure supplement 4.** Reconstruction of tubular structures, nuclei and cells.

**Figure supplement 5.** Generation of realistic 3D images of liver tissue.

**Figure supplement 6.** Benchmark of images to evaluate 3D reconstructions of dense tissue.

**Figure supplement 7.** Model validation: Evaluation of the accuracy of our pipeline for the 3D reconstruction of dense tissue.

## Cell classification and reconstruction

Generating geometrical models of tissues requires the proper recognition of different cell types. A previous automated classification system discriminated hepatocytes from non-parenchymal cells in 2D human liver images with a 97.8% accuracy (*O'Gorman et al., 1985*). However, the automatic classification of non-parenchymal cells in 3D liver tissue is more challenging. Given their importance in physiology and disease (*Bouwens et al., 1992*; *Kmiec, 2001*; *Malik et al., 2002*) and the limitation on the number of fluorescent markers that can be simultaneously imaged, we designed an algorithm to automatically classify different cell types in the tissue based on nuclear morphological features. We chose two deterministic supervised classifiers, linear discriminant analysis (LDA) and Bayesian network classifier (BNC). LDA, also known as Fisher LDA (*Fisher, 1936*), is a fundamental and widely used technique to classify data into several mutually exclusive groups (*Duda et al., 2001*). It has been successfully applied for nuclei discrimination in microscopy images (*Huisman et al., 2007*; *Lin et al., 2007*). On the other hand, BNCs are more recently developed classifiers which not only show good performance but also allow for probabilistic classification. In

addition, BNCs reveal the hierarchy of parameters used for the classification (*Friedman et al., 1997*), which may provide insights into underlying biological processes.

As input for the classifiers, we manually built a training set of 2301 nuclei using specific cellular markers (*Figure 3—figure supplement 2A*) and computed for each nucleus a profile of 74 parameters (*Table 1*) describing nuclei morphology, texture and localization relative to sinusoids and cell borders (density of actin in vicinity of nuclei) (see 'Methods'). For the LDA, the parameters were ranked using Fisher score (*Duda et al., 2001*), and the most relevant ones were selected based on the classification accuracy (*Figure 3—figure supplement 2B* and 'Methods'). Independently, the most relevant parameters were selected on the basis of Bayesian network structure reconstruction (*Friedman et al., 1999*) (*Figure 3—figure supplement 2C*).

The performance of the classifiers was measured using the leave-one-out cross-validation method on the training set. Both classifiers recognized hepatocytes with ~100% accuracy, thus further improving the previous performance (*O'Gorman et al., 1985*). The overall cell-type classification yielded 95.4% and 92.6% accuracy for the LDA and BNC, respectively. Although discriminating non-parenchymal cells is difficult even for a person skilled in the art, our algorithms achieved accuracy higher than 90%. The predictive performance of the classifiers is shown in *Figure 3—figure supplement 3A,B*. As expected, the first largest population of cells corresponds to hepatocytes (44.6% ± 2.7%, mean ± SEM) followed by sinusoidal endothelial cells (29.8% ± 2.5%). Surprisingly, we found important quantitative differences for Kupffer and stellate cells. The percentage of Kupffer cells (8.7% ± 0.7%) was lower than that of stellate cells (11.2% ± 1.0%), against previous estimates on 2D images (*Baratta et al., 2009*). The percentage of other cells was 5.7% ± 0.8%. A 3D visualization of the localization of the nuclei of the different cell types is shown in *Figure 3—figure supplement 3C–F*.

Finally, cells were segmented by expansion of the active mesh from the nuclei to the cell surface. The expansion was either limited to the cell cortex (i.e. the maximum density of actin) or to contacts with neighbouring cells or tubular structures (*Figure 3E*). The active mesh expansion was parameterized by inner pressure and mesh rigidity. However, this algorithm over-segmented bi-nucleated cells into two cells with a single nucleus. Therefore, we used phalloidin intensity and nucleus-to-nucleus distance to recognize over segmented multinuclear cells and merge them. A manual check of segmentation of 2559 cells revealed only ~2% error for hepatocyte segmentation that is a further improvement of the state-of-the-art achievements by voxel-based segmentation methods (*Mosaliganti et al., 2012*). The results of the segmentation of all imaged cellular and subcellular structures in the liver tissue (i.e. cells, nuclei, sinusoidal and BC networks) are presented in *Figure 3E*, *Figure 3—figure supplement 4*, and *Videos 2* and *3*.

## Model validation

To evaluate the performance of the pipeline for the reconstruction of dense tissues, we generated a benchmark comprising a set of realistic 3D images of liver tissue. Each synthetic image consisted of four channels for the main structures forming the tissue, that is, cell nuclei, cell borders, sinusoidal and BC networks. We first generated 3D models of liver tissue based on experimental data (see 'Methods'). The benchmark models had levels of complexity similar to that of the real tissue (*Figure 3—figure supplement 5,6*). Second, we imposed uneven staining to the models in order to resemble the experimental data. Third, the artificial microscopy images were simulated by convolving the models according to the 3D confocal microscope point spread function (PSF) (*Nasse et al., 2007*; *Nasse and Woehl, 2010*) and adding z-dependent Poisson noise. The resulting benchmark image statistics were similar to those from the images acquired in our experimental setup (see 'Methods') (*Figure 3—figure supplement 5*). Given their general usefulness for testing image analysis software, the benchmark images and models are provided as supplementary material (*Supplementary file 1*, *Morales-Navarrete et al., 2016*). Finally, we applied our 3D tissue reconstruction pipeline to the benchmark images and quantified the accuracy of the reconstructed models using the precision-sensitivity framework (*Powers, 2011*). The overall quality was expressed as F-score, the harmonic mean between precision and sensitivity. The benchmark tests were performed in three sets of images with different signal-to-noise ratio (10:1, 4:1, 2:1). For tubular structures, we achieved average (over the different noise level sets) F-scores of 0.90 ± 0.04 and 0.73 ± 0.06 for sinusoidal and BC networks, respectively. In the case of the nuclei and cell segmentation, we found average F-scores 0.91 ± 0.03 and 0.92 ± 0.03, respectively. The detailed quantifications

**Table 1.** List of the 74 parameters calculated for the nuclei classification.

| Parameter | F-score | Parameter | F-score |
|---|---|---|---|
| FLK1 surface intensity 1 vx | 4.802 | Mean radius | 0.920 |
| FLK1 surface intensity 0 vx | 4.737 | FLK1 KURT | 0.915 |
| FLK1 mean | 4.674 | MB Frac Dim | 0.904 |
| FLK1 surface intensity 2 vx | 4.570 | Log Lac2 | 0.885 |
| FLK1 surface intensity 3 vx | 4.100 | HF2 | 0.833 |
| Phallo surface intensity 2 vx | 3.477 | HF13 | 0.825 |
| FLK1 surface intensity 4 vx | 3.453 | HF3 | 0.817 |
| Phallo surface intensity 1 vx | 3.430 | Phallo surface intensity 9 vx | 0.787 |
| FLK1 SKEW | 3.351 | Surface area | 0.768 |
| Phallo surface intensity 3 vx | 3.253 | Log lac 3 | 0.718 |
| Phallo surface intensity 0 vx | 3.236 | Radius variance | 0.669 |
| Norm lac 3 | 2.930 | Volume | 0.668 |
| Norm lac 2 | 2.913 | BC Frac Dim | 0.649 |
| FLK1 surface intensity 5 vx | 2.857 | Log lac 4 | 0.612 |
| Norm lac 4 | 2.847 | Phallo surface intensity 10 vx | 0.554 |
| Phallo surface intensity 4 vx | 2.838 | Log lac 5 | 0.536 |
| Norm lac 5 | 2.753 | Sphericity | 0.423 |
| Phallo surface intensity 5 vx | 2.347 | HF7 | 0.408 |
| FLK1 surface intensity 6 vx | 2.310 | Shape index | 0.402 |
| HF9 | 2.141 | Lacunarity 1 | 0.381 |
| FLK1 surface intensity 7 vx | 1.893 | b/c | 0.342 |
| Phallo surface intensity 6 vx | 1.868 | Lacunarity 2 | 0.333 |
| HF5 | 1.575 | Lacunarity 3 | 0.309 |
| HF8 | 1.554 | Lacunarity 4 | 0.295 |
| FLK1 surface intensity 8 vx | 1.552 | HF4 | 0.287 |
| HF11 | 1.471 | Lacunarity 5 | 0.285 |
| Phallo surface intensity 7 vx | 1.444 | HF12 | 0.153 |
| a/c | 1.406 | DAPI Sd | 0.123 |
| Log lac 1 | 1.287 | DAPI gradient surface | 0.094 |
| FLK1 surface intensity 9 vx | 1.265 | Log norm lac 2 | 0.087 |
| HF6 | 1.158 | CVM | 0.076 |
| Phallo surface intensity 8 vx | 1.084 | Log norm lac 3 | 0.062 |
| FLK1 surface intensity 10 vx | 1.018 | Log norm lac 4 | 0.045 |
| HF1 | 0.978 | DAPI SKEW | 0.035 |
| FLK1 Sd | 0.942 | Log norm lac 5 | 0.033 |
| HF10 | 0.939 | DAPI mean | 0.029 |
| a/b | 0.937 | DAPI KURT | 0.022 |

Note: The parameters are sorted based on their Fisher score, which is a measure of the discriminative power of the parameter.

are shown in *Figure 3—figure supplement 7A–L*. Additionally, we measured morphometric parameters of the reconstructed structures such as the average radius of the tubular structures (BC and sinusoidal networks) and cell volumes. We obtained values of 2.72 ± 0.13 µm (ground truth value = 3.0 µm) and 0.58 ± 0.05 µm (ground truth value = 0.5 µm) for sinusoidal and BC networks,



**Video 2.** Reconstruction of all imaged structures in a high-resolution image. A 2x2 stitched (~ 400 µm × 400 µm × 100 µm) high-resolution image (0.3 µm × 0.3 µm × 0.3 µm per voxel) was used. First, the reconstruction of the large vessels, that is, central vein (CV) (cyan), portal vein (PV) (orange) and bile duct (green) are shown. Then, raw images and the corresponding reconstructed objects of the different structures are shown sequentially: sinusoids (magenta), BC (green), nuclei (random colours) and cells (random colours). Additionally, central lines are shown for the tubular structures. Finally, all segmented structures are shown. This video provides a complete over view of the reconstructed objects in a typical high-resolution image.

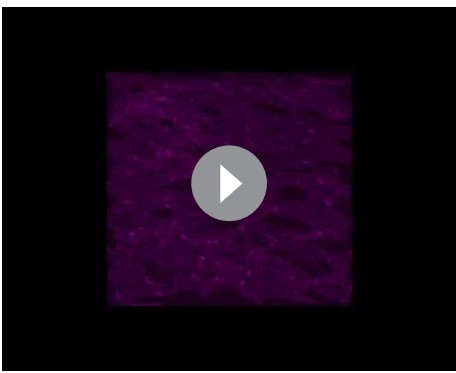

**Video 3.** Detailed reconstruction of all imaged structures in a high-resolution image. In order to highlight the details of the reconstruction of small structures [e.g. nuclei, bile canalicular (BC) network, etc.], a video of a small, cropped (~125 µm × 125 µm × 75 µm) high-resolution image (3 µm × 0.3 µm × 0.3 µm per voxel) was generated. Similarly to *Video 2*, the raw image and the corresponding reconstructed structures of sinusoids (magenta), BC (green), nuclei (random colours) and cells (random colours) are shown sequentially.

respectively (*Figure 3—figure supplement 7M, N*). The average error for cell volume estimation was found to be 5.17% ± 1.97% (*Figure 3—figure supplement 7O*). The benchmark experiments showed high accuracy for the reconstruction of the 'ground truth' models of all the morphologically different structures forming the liver tissue (*Figure 3—figure supplement 7*).

## New insights into liver tissue organization from the geometrical model

Next, we applied our software to quantitatively analyse the geometric features of liver tissue from three adult mice. Geometric features have important implications, for example, for the development of models of fluid exchange between blood and hepatocytes (*Wisse et al., 1985*). A critical parameter for blood flux models is the radius of sinusoids. We measured a radius of 4.0 ± 1.1 µm, a value close to quantifications by electron microscopy (EM) analysis (*Wisse et al., 1985*; *Oda et al., 2003*; *McCuskey, 2008*). In the sinusoidal networks, we determined the angles between two branching arms to be 111.6° ± 12.37° (*Figure 4—figure supplement 1B*), against previous estimates (*Hammad et al., 2014*). Moreover, the values for the BC network are similar to the sinusoidal network (110.36° ± 9.85°, *Figure 4—figure supplement 1B*). Additionally, we provided new geometric information such as the cardinality of the branching nodes (*Figure 4—figure supplement 1C*).

Recent studies on the morphometric parameters of the liver tissue (*Hammad et al., 2014*; *Friebel et al., 2015*) provided either average values or limited data measurements of the hepatocytes volume, omitting information on their heterogeneity. We could not only perform accurate measurements of hepatocytes volumes and poly-nucleation, but also correlate them with polyploidy and spatial localization within the tissue. Interestingly, we found a multi-modal distribution of hepatocyte volumes (*Figure 4A*) in line with measurements on isolated hepatocytes (*Martin et al., 2002*). A trivial explanation is that it reflects the presence of mono- and bi-nucleated hepatocytes. However, we found that this was not the case. The distribution of volumes of both mono- and bi-nucleated hepatocytes can be independently described by a mixture of two populations with mean volumes 3126 ± 1302 µm³ (~14% of cells) and 5313 ± 1175 µm³ (~10% of cells), and 5678 ± 1176 µm³ (~45% of

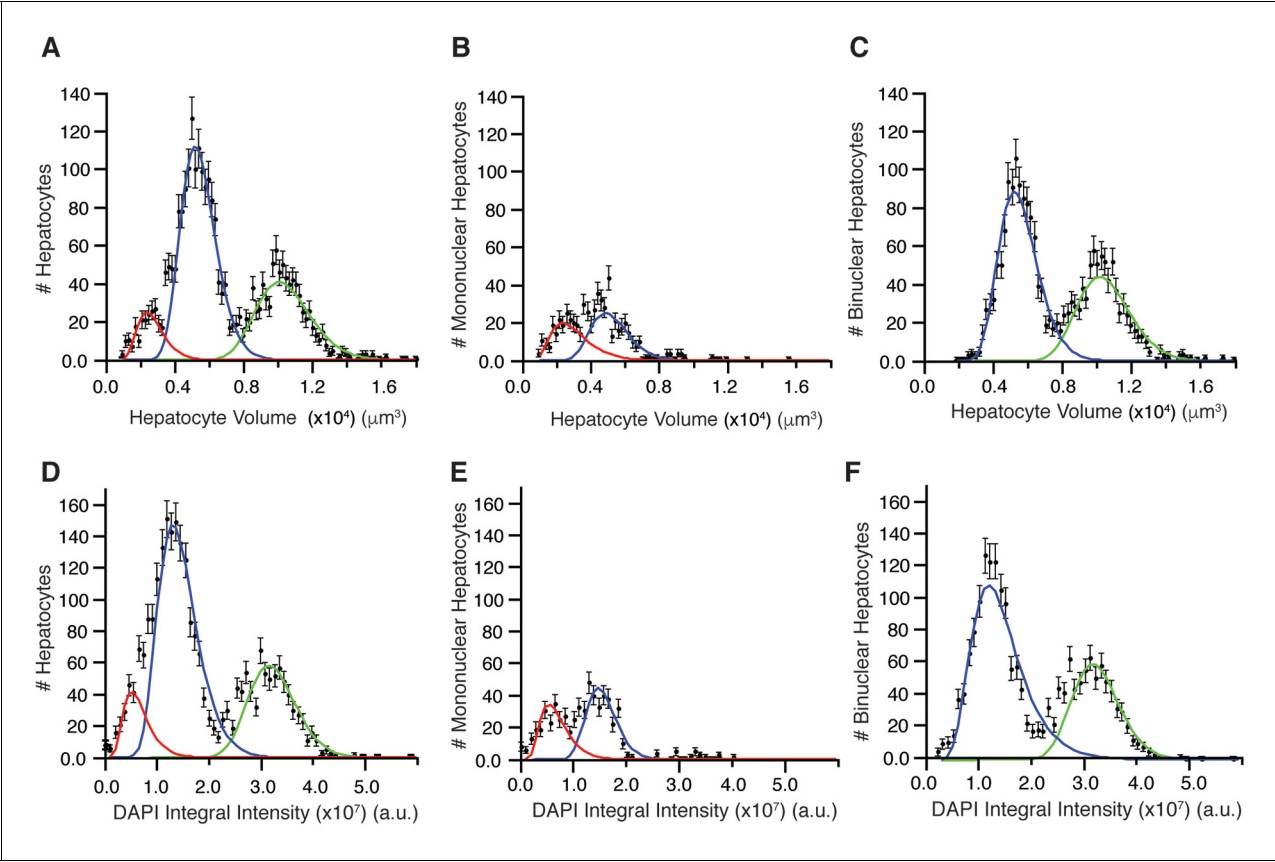

**Figure 4.** Distribution of hepatocyte volumes and DAPI integral intensity per cell for all hepatocytes (A, B) and separated by number of nuclei (B, C and E, F). Whereas experimental data are shown by dots, the log-normal components fitted to data are shown by solid lines. (A) Cell volume distribution of all hepatocytes. (B, C) Cell volume distribution obtained for mono and bi-nucleated hepatocytes, respectively. (D) Distribution of DAPI integral intensity (proportional to the content of DNA) of all hepatocytes. (E, F) Distributions of DAPI integral intensity obtained for mono and bi-nucleated hepatocytes, respectively. The analysis was performed on 2559 hepatocytes (excluding boundary cells) from three adult mice.

The following figure supplements are available for figure 4:

**Figure supplement 1.** Morphometric features of the sinusoidal and bile canalicular (BC) networks.

**Figure supplement 2.** (A, B) DAPI integral intensity normalization.

cells) and 10606 ± 1532 $\mu m^3$ (~30% of cells), respectively (*Figure 4B,C*). Hence, surprisingly, although the bi-nucleated hepatocytes are assumed to be larger than the mono-nucleated, we found that a population of mono-nucleated hepatocytes can have a volume that does not differ from that of bi-nucleated hepatocytes (*Figure 4A–C*).

Having found such a peculiar size distribution of bi-nucleated hepatocytes, we measured the total content of DNA per nucleus in every cell sub-population as the integral intensity of DAPI (*Coleman et al., 1981*; *Xing and Lawrence, 1991*; *Dmitrieva et al., 2011*; *Zhao and Darzynkiewicz, 2013*) (see 'Methods'). The resulting distribution (*Figure 4D*) shows three well-separated peaks. These presumably correspond to the 2n (diploid nuclei), 4n and 8n (polyploid nuclei) DNA content previously reported (*Guidotti et al., 2003*; *Martin et al., 2002*) (note that this analysis does not resolve the aneuploidy of specific chromosomes (*Faggioli et al., 2011*)).

Next, we asked how the nuclei are distributed between the mono- and bi-nucleated cell populations. Interestingly, in the small bi-nucleated hepatocytes (volume < 8000 $\mu m^3$) both nuclei had 2n DNA content, whereas in the large hepatocytes (volume > 8000 $\mu m^3$) both had 4n DNA content. Almost no bi-nuclear hepatocytes (<1.0%) with different amount of DNA per nucleus (e.g. one nucleus with 2n and one with 4n) were observed (*Figure 4—figure supplement 2C,D*). These results

suggest that the hepatocyte volume does not depend on the number of nuclei but rather on their polyploidy, in agreement with previous reports (*Miyaoka and Miyajima, 2013*). Therefore, we classified hepatocytes with respect to number of nuclei, volume and DNA content using a hierarchical cluster algorithm. We identified seven populations, namely 2n, 4n, 8n, 16n for mono-nuclear and 2×2n, 2×4n, 2×8n for bi-nuclear hepatocytes (*Figure 4—figure supplement 2E,F*). Four populations (mono-nucleated 2n and 4n, and bi-nucleated 2×2n and 2×4n) were major, representing around 97% of all hepatocytes.

The reports on the spatial distribution of polyploid hepatocytes are controversial (*Gentric and Desdouets, 2014*). Whereas some suggest that periportal hepatocytes show a lower polyploidy than the perivenous ones (*Gandillet et al., 2003*; *Asahina et al., 2006*), others suggest that both regions have similar polyploid compositions (*Margall-Ducos et al., 2007*; *Pandit et al., 2012*). These discrepancies prompted us to analyse the spatial distribution of mono- and bi-nucleated hepatocytes within the lobule. We particularly analysed the largest populations of hepatocytes, 2n, 4n, 2×2n and 2×4n. Strikingly, we found a pronounced zonation in their localization. Whereas the 2n mono-nucleated were enriched in the PC and PV regions, mono-nucleated 4n showed a homogeneous distribution between PV and PC regions (*Figure 5*). The 2×2n bi-nucleated hepatocytes have a similar pattern as the 2n mono-nucleated (highly enriched in the CV and PV regions), but the density of 2×4n bi-nucleated was lower in those regions and increased in the middle region (*Figure 5*). As far as we know, this is the first time that polyploidy and poly-nuclearity are found to be zonated and follow a specific pattern. These findings have important implications for both the structural organization of liver tissue and its proliferating and metabolic activities.

## Application of the pipeline to lung and kidney tissue

To test the general applicability of the pipeline as well as the robustness of our algorithms, we applied it to two morphologically distinct tissues, lung and kidney. Lung and kidney sections were stained for nuclei (DAPI) and the cell cortex (F-actin by phalloidin). Kidney samples were additionally stained for the apical (CD13) and basal (fibronectin and laminin) cell surface. The pipeline allowed us to generate geometrical reconstructions of the tissues (*Figure 6* and *Videos 4* and *5*, respectively) without fine-tuning of the parameters. As proof of principle, we extracted some statistics of the most relevant structures from each tissue. Structural information from both relatively large structures like alveoli in lung or glomerulus in kidney, and smaller ones like cells and nuclei were extracted from the geometrical models. *Figure 6—figure supplement 1,2* show the statistical distributions of some interesting tissue features, such as cell volume and elongation, number of neighbouring cells, etc. Information about the spatial organization of the alveolar cells (i.e. their localization relative to the alveoli) in the lung was extracted as well.

For example, in the lung, we found that the alveolar cells constitute around 19% of the volume, consistent with previous measurements (*Irvin and Bates, 2003*). In the kidney, we found that proximal tubule cells have larger volumes than distal tubule cells (*Figure 6—figure supplement 2*), also in agreement with previous studies (*Nyengaard et al., 1993*; *Rasch and Dørup, 1997*). Altogether, the new data show that our pipeline is versatile and able to reconstruct geometrical models of tissues with fairly different architectures.

## Discussion

We developed a versatile pipeline that combines new algorithms with established ones aimed to reconstruct geometrical models of dense tissues from confocal microscopy images acquired at different levels of resolution. Our pipeline is implemented in a freely available platform designed to address unmet computational needs. Despite many efforts, the reconstruction of digital geometrical models of tissues suffers from critical bottlenecks such as lack of automation, limited accuracy and low throughput analysis (*Peng et al., 2010*). The platform developed here overcomes such bottlenecks in that it (1) achieves high accuracy of geometric reconstruction, (2) can process large volumes of imaged tissue, for example, a full liver lobule, (3) increases the image analysis performance to such an extent that the model reconstruction time is shorter than the biological experimental time and compatible with middle-throughput (this is achieved by combining the computational efficiency of C++ with the CPU/GPU multi-threading capabilities), (4) can be run on a regular PC and (5) provides a flexible tool for constructing image processing pipelines that are tuneable for specific tissue

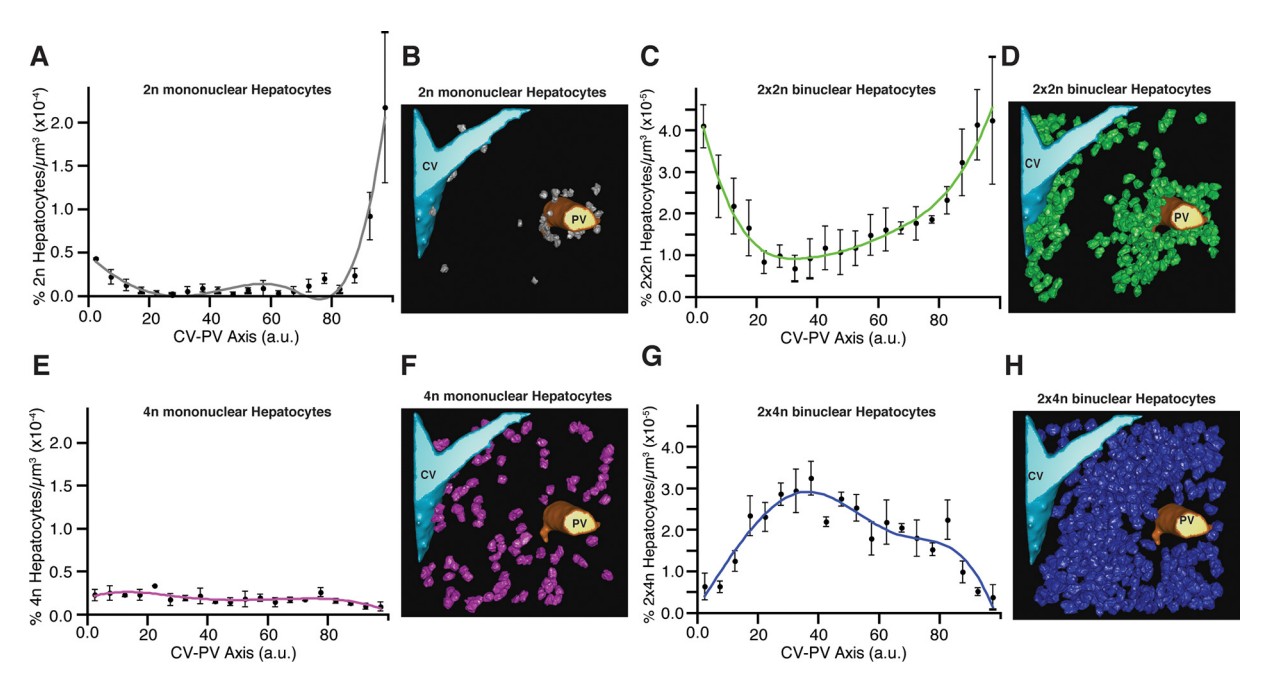

**Figure 5.** Relative density of different sub-populations of hepatocytes as function of central vein (CV)-portal vein (PV) axis coordinate. (**A, C, E, G**) Relative density of 2n mono-nucleated, 2x×2n bi-nucleated, 4n mono-nucleated, 2x×4n bi-nucleated hepatocytes, respectively. (**B, D, F, H**) 3D visualization of the corresponding sub-populations of hepatocytes. The analysis was performed on 2559 hepatocytes (excluding boundary cells) from three adult mice. The CV-PV axis is determined by the coordinate χ, which describes the position of a point relative to the closest CV and PV.

$\chi = 50 \times \left( \frac{|D - d_{pv}| - |D - d_{cv}|}{D} + 1 \right)$, where $d_{cv}$ and $d_{pv}$ are the distances to the closest CV and PV respectively, and $D$ is the CV-PV distance. χ takes values

between 0 and 100, where 0 and 100 represents a localization at the CV and PV surfaces, respectively.

and imaging conditions. For the automatic recognition of different cell types, we included morphological classifiers into the software. The user-friendly pipeline assembly mechanism allows adjusting the platform for specific tissue analysis demands. The newly developed algorithms both increase the quality of the results (e.g. 3D image de-noising, LME method, active mesh tuning, cell classification) and deal with problems for which there appears currently to be no real good solutions available (e.g. correction of tissue deformation and combination of individual sections in the case of partial tissue removal) (*Figure 1—figure supplement 1*). Our platform is implemented as stand-alone free to download software (http://motiontracking.mpi-cbg.de). Furthermore, we created a benchmark of realistic images (with the underlying ground truth model) for the evaluation of 3D segmentation algorithms in biological images (*Supplementary file 1*, *Morales-Navarrete et al., 2016*).

To test its efficacy, we applied it towards the generation of a multi-resolution geometrical model of liver tissue. The resulting model was used to extract quantitative measurements of various features of liver tissue organization, such as radius, branching angles and cardinality of the sinusoidal and BC networks, and to recognize different cell types based on their morphological parameters. Our analysis revealed an unexpected zonation pattern of hepatocytes with different size, nuclei and DNA content within the liver lobule. Furthermore, we extended the analysis to two additional tissues, lung and kidney, demonstrating the general applicability and robustness of our platform.

In building our pipeline, we spent considerable effort to improve the accuracy of the measurements of cell and tissue parameters and preserve their contextual information. The new algorithms allow correcting major defects originating from tissue sectioning, improve the segmentation of cellular, subcellular and tissue-level structures, and extract morphological features and distributions in space. A major limiting factor in the development of a comprehensive geometrical model is the trade-off between imaging large volumes of samples to gain a view of the overall tissue architecture and imaging at high-resolution to achieve an accurate description of the structures at the limit of resolution of the light microscope, for example, the apical surface of hepatocytes forming the

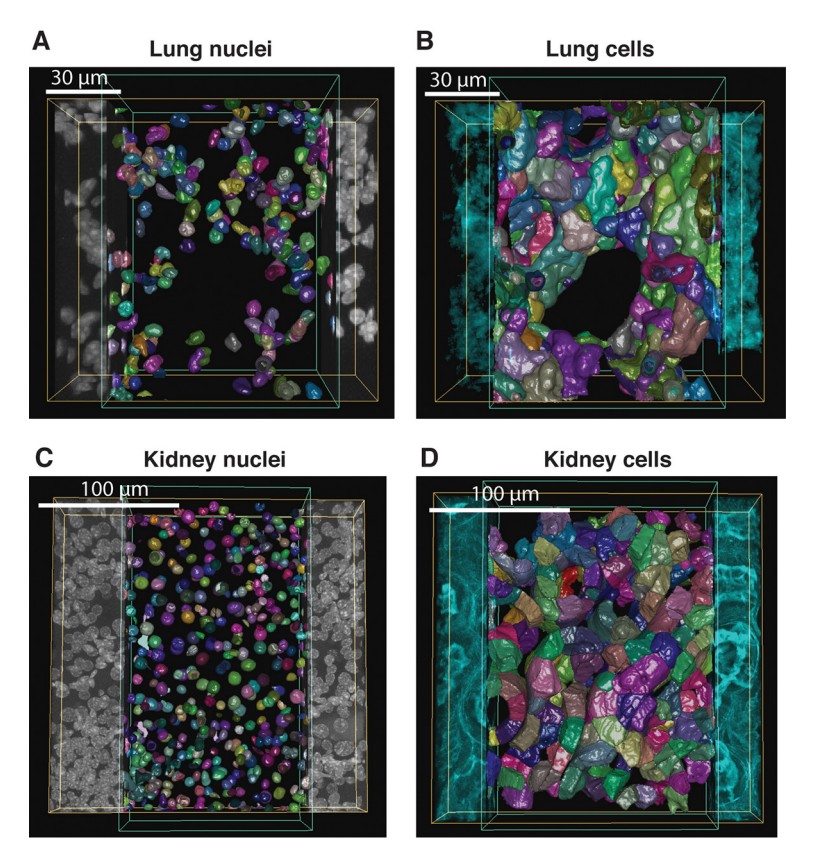

**Figure 6.** Reconstruction of geometrical models of lung and kidney tissues. 3D representation of the different structures segmented in each tissue: (A, C) nuclei and (B, D) cells in the lung and kidney tissues, respectively. The triangle meshes are drawn inside the inner box and the raw images outside.

The following figure supplements are available for figure 6:

**Figure supplement 1.** Morphometric features of lung tissue.

**Figure supplement 2.** Morphometric features of kidney tissue.

BC network. We solved this problem by imaging the tissue at low-resolution and registering within it the parts of tissue (the PV-PC area in the case of the liver lobule) imaged at high resolution. In this way, the measured morphological features (e.g. BC) and parameters (e.g. cell size) are embedded in their proper context of tissue architecture. For example, the hepatocyte volume is a parameter that has little value as average without considering the distribution of parameter values in the lobule (*Figure 5*). In general, the diversity of geometric features of the cells within the liver lobule could provide new insights into the regulation of metabolic zonation (see below).

Our nuclei reconstruction approach achieved accuracy higher than 90%. As shown in *Figure 3—figure supplement 1K*, the major source of errors is over-segmented nuclei. Additional steps to improve nuclei reconstruction, such as the region-merging algorithm (*Chittajallu et al., 2015*) to correct for over-segmentation, could reduce such errors. Even though our cell segmentation method proved able to identify and reconstruct cells with high accuracy, in a few cases (~2%), binuclear cells were mistaken for two separate cells due to weak staining of the cell cortex. Therefore, implementation of additional methods for enhancing the staining of the cell surface, such as the anisotropic plate diffusion filters (*Mosaliganti et al., 2010*; *2012*), could help reduce further the over-segmentation of multi-nuclear cells.

The active mesh tuning allowed improving the accuracy of segmentation of the BC and sinusoidal networks. This is important since the accuracy of a geometrical model is indispensable for the

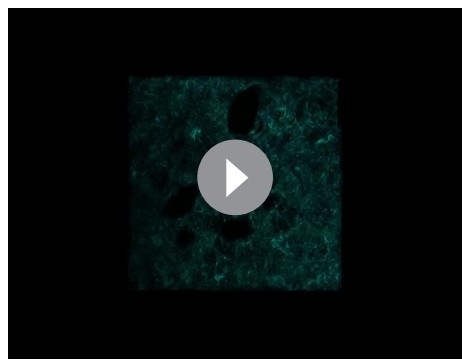

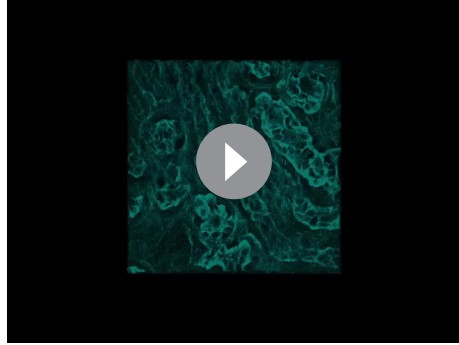

**Video 4.** 3D reconstruction of lung tissue. Nuclei and cells reconstructed from a high-resolution image (~220 µm × 220 µm × 80 µm). First, the raw images of the cell cortex (F-actin by phalloidin) and nuclei (DAPI) staining are displayed. Then, the reconstruction of the nuclei (random colours) and the cells (random colours) are shown.

**Video 5.** 3D reconstruction of kidney tissue. Nuclei and cells reconstructed from a high-resolution image (~220 µm × 220 µm × 80 µm). First, the raw images of the cell cortex (F-actin by phalloidin) and nuclei (DAPI) staining are displayed. Then, the reconstruction of the nuclei (random colours) and the cells (random colours) are shown.

development of predictive models of tissue function. For example, a model of blood flow through the sinusoidal network and exchange with hepatocytes via the space of Disse (*Ohtani and Ohtani, 2008*; *Wisse et al., 1985*) critically depends on the estimation of the sinusoid diameter. An overestimation of the sinusoidal tube radius would have major consequences for the predictions of blood cells flow through the sinusoidal network. Our geometrical model yielded a diameter of the sinusoidal-walled tube equal to the typical size of erythrocytes and lymphocytes. Therefore, it supports the model of active exchange of blood serum and lymph in the space of Disse, whereby blood flux propels cells through the sinusoids causing waves of capillary walls deformation (*McCuskey, 2008*; *Wisse et al., 1985*). The active mesh tuning algorithm yielded a distribution of the radius of sinusoid capillaries with a mean value that was 20% lower (*Figure 4—figure supplement 1A*) than previously estimated by similar approaches (*Hammad et al., 2014*; *Hoehme et al., 2010*), but in line with the values reported by EM (*Wisse et al., 1985*). The reconstruction also revealed a large difference with the previously reported angles between two arms of branching sinusoids (112° vs. 32°, *Figure 4—figure supplement 1B*). Moreover, the geometrical model provides correct values for other sinusoidal network parameters such as number of intersection nodes per mm$^3$ ($8.3 \times 10^4 \pm 1.9 \times 10^4$) and network length per mm$^3$ ($3.1 \times 10^6 \pm 0.3 \times 10^6$ µm), which appear to have been overestimated in a recent report (*Hammad et al., 2014*) (see 'Methods'). The discrepancy between our geometrical model and others (*Hoehme et al., 2010*; *Hammad et al., 2014*) could be due to differences in image processing and/or experimental procedures (tissue fixation, image acquisition, etc.). One possible explanation for this discrepancy is that our platform applies the active mesh approach to the segmentation of structures on different scales (from the apical surface of hepatocytes forming the BC to cells) and this may yield a more precise geometrical reconstruction in comparison with voxel-based methods (*Figure 3A*).

For the marker-less cell-type recognition, we compared two approaches, the classical LDA and the more recent BNC, applied to nuclei morphology. The accuracy of both approaches was comparable, reaching higher than 99% efficiency for hepatocyte recognition and about 92–95% for all cell types. The latter value is highly significant since the distinction between stellate and sinusoid endothelial cells in the absence of specific markers is challenging even for a skilled person. The analysis of parameters that were mostly informative for cell type discrimination yielded some unexpected results. Although nuclear size and roundness were traditionally considered a priori as the most relevant parameters to discriminate hepatocytes from non-parenchymal cells (*Baratta et al., 2009*; *O'Gorman et al., 1985*), we found that they are less informative than the parameters related to nuclear texture (e.g. moments of lacunarity). The analysis of parameters relevant for cell classification can shed light on the differences in cell morphology that are difficult to grasp by the naked eye. The accurate active mesh-based cell shape estimation led to well-separated peaks of cell volume

distribution (*Figure 4A–C*), which failed to be discriminated by approximation through Voronoi tessellation (*Bock et al., 2010*) (data not shown).

The analysis of liver tissue using our software platform revealed some unexpected biological findings. It is well established that hepatocytes are heterogeneous in size, number of nuclei (mono and bi-nucleated cells) and DNA content (polyploidy). However, we observed that these features are not randomly distributed but follow a specific zonation pattern within the liver lobule. Surprisingly, the mono-nucleated 2n and bi-nucleated 2×2n hepatocytes were enriched in the CV and PV regions, whereas bi-nucleated 2×4n were more frequent in the middle region. This particular distribution suggests that polyploidy is spatially regulated and follows a gradient between CV and PV. Zonation of metabolic activities in the liver is well known, but zonation of mono- and bi-nucleated cells and total DNA content (polyploidy) remains controversial. The spatial distribution of hepatocytes according to their ploidy in the CV-PV axes correlates with the metabolic zonation. This correlation suggests a possible role of polyploidy in regulating hepatocyte functions in the liver lobule. Interestingly, two unique populations of cells with stem cell-like properties and the capacity to repopulate the liver have been recently identified (*Ray, 2015*; *Wang et al., 2015*; *Font-Burgada et al., 2015*). One population located close to the CV, which has been implicated in homeostatic hepatocyte renewal (*Wang et al., 2015*), coincides with the mono-nucleated 2n cells we identified. The other population of hepatocytes located near the PV, which was found to repopulate the liver after injury (*Font-Burgada et al., 2015*), could correspond to the low ploidy cells (2n and 2×2n) we observed. These results inspire future studies aimed at exploring the mechanisms underlying regulation of mono- versus bi-nuclearity and polyploidy in the context of liver tissue structure, function and regeneration (*Zaret, 2015*; *Ray, 2015*). In this context, the accurate digital geometrical model of tissue is a valuable resource.

Geometrical models provide the means of extracting structural information as a precondition for the development of functional models of tissues. They can be a tool for acquiring accurate quantitative measurements of morphological features and, as such, have the potential of uncovering the fundamental rules underlying tissue organization. In addition, the measurement of specific parameters, such as BC and sinusoid diameters, network cardinality, cell volume and shape, etc., can serve as diagnostic markers of early stages of tissue dysfunction/repairing, thus providing new tools for clinical research and drug development.

## Methods

### Mice and ethics statement

Six- to nine-week-old C57BL/6JOlaHsd mice were purchased from Charles River Laboratory. All animal studies were conducted in accordance with German animal welfare legislation and in strict pathogen-free conditions in the animal facility of the Max Planck Institute of Molecular Cell Biology and Genetics, Dresden, Germany. Protocols were approved by the Institutional Animal Welfare Officer (Tierschutzbeauftragter) and all necessary licenses were obtained from the regional Ethical Commission for Animal Experimentation of Dresden, Germany (Tierversuchskommission, Landesdirektion Dresden) (license number: AZ 24-9168.24-9/2012-1, AZ 24-9168.11-9/2012-3).

### BFBD algorithm for de-noising images of fluorescent microscopy

We took advantage of the fact that point-spread-function of confocal microscopes is strongly elongated in z-axis and developed a new de-noising algorithm based on the linear approximation of the image background intensity in the z-direction. Since confocal microscopy images are photon-limited and therefore obey Poisson statistics, we first found the parameters $\alpha$ and $\beta$ that convert the photon counts ($N$) into the intensity ($I$) units, such that:

$$\langle I \rangle = \alpha \langle N \rangle + \beta$$

where the operator $\langle . \rangle$ represents the average, $\alpha$ is the conversion coefficient from number of photons to intensity values and $\beta$ is the offset of the microscope digitization system (dark current).

For this, we calculated the variance of the intensities between sequential optical z-sections for each X–Y pixel and binned them according to the pixel intensities. Then, the mean variance was calculated within each bin and, as a result, the dependency of mean variance upon the intensities was

found (*Figure 1—figure supplement 2G*). This dependency was found to be linear, as expected for a Poisson noise model:

$$V(I) = \alpha^2 \langle N \rangle = \alpha(\langle I \rangle - \beta)$$

where *V(I)* is the variance for each intensity level $\langle I \rangle$.

Moreover, when thick 3D tissue samples are imaged, it is required to use different laser intensity and microscope gain. This results in an increase of the intensity scaling factor $\alpha$ with the image depth. Therefore, we calculated the Poisson noise model for different image depths (z-direction) and then, we used $\alpha$ and $\beta$ to estimate the variance for every pixel.

After that, we estimated the background intensity of every pixel. Briefly, for each pixel a set of sequential intensities in z-direction was extracted (*Figure 1—figure supplement 2H*, left). Then, the intensities were fitted by a straight line using the outlier-tolerant algorithm described in (*Sivia, 1996*) (*Figure 1—figure supplement 2H*, right). The prediction of the straight line was considered as the background intensity, and the difference between the measured intensity and background was considered as candidate foreground intensity. The candidate foreground intensities below a defined threshold (expressed in variance units) were excluded. Finally, the background was added to the foreground to form the de-noised image.

To evaluate the performance of our algorithm, we applied it to a set of three artificial images of BC from our benchmark (2:1 signal-to-noise ratio). Additionally, we applied other methods such as median filtering, Gauss low-pass filtering and anisotropic diffusion, 'pure denoise' (PD) (*Luisier et al., 2010*) and 'edge preserving de-noising and smoothing' (EPDS) (*Beck and Teboulle, 2009*) for comparison. The performance of each method was quantitatively evaluated using the metrics mean square error (MSE) and coefficient of correlation (CoC), defined as follows:

$$\mathrm{MSE} = \frac{\sum_{i \in \Omega}(I_i - I_i^*)^2}{|\Omega|}$$

$$\mathrm{CoC} = \frac{\sum_{i \in \Omega}(I_i - \langle I \rangle) \cdot (I_i^* - \langle I^* \rangle)}{(\sum_{i \in \Omega}(I_i - \langle I \rangle)^2 \cdot \sum_{i \in \Omega}(I_i^* - \langle I^* \rangle)^2)^{1/2}}$$

where $\Omega$ is the region of interest in the image, $I_i$ and $I_i^*$ are the intensities at voxel *i* of the de-noised and noise-free (ground truth) images respectively, $\langle I \rangle$ and $\langle I^* \rangle$ are the mean intensities of the de-noised and noise-free images, respectively. We calculated the MSE and CoC over the whole images (global) as well as in the vicinity of the objects (*Figure 1—figure supplement 3A*). For PD and EPDS, we selected the best parameters for their performance before the comparison (*Figure 1—figure supplement 3B,C*). The results of our quantifications are shown in *Figure 1—figure supplement 4–5*.

## Methods for the reconstruction of 3D multi-scale images

### Reconstruction of physical sections

To image large and complex tissue structures such as the liver lobule, we generated a grid of partially overlapping low-resolution 3D images (stacks) for each individual tissue section. We applied an image mosaicking procedure to merge the stacks into a single 3D image of the section (*Figure 2A* and *Figure 2—figure supplement 1A*). Our merging procedure adopts a standard approach to maximize the sum of cross-correlations calculated for pairs of neighbouring tiles in the grid. The input data set for the reconstruction of physical sections was composed of N-by-M grids of partially overlapping 3D images (Z-stacks) (*Figure 2—figure supplement 1A*). It is assumed that an approximation of their overlapping area is known and that transitional image registration is sufficient for reconstruction purposes.

Let $(Z_{x,y}Z_{x',y'})$ be a pair of neighbour images located within the grid ($0 \leq x < N$, $0 \leq y < M$, $0 \leq x' < N$, $0 \leq y' < M$, $|x - x'| = 1 \curlyvee |y - y'| = 1$), and $C_{x,y,x',y}(i,j,k)$ the cross-correlation of their overlapping areas. The quality of their local alignment for a given shift (*i,j,k*) is measured by the corresponding value of the cross-correlation $C_{x,y,x',y}(i,j,k)$. The goal of the reconstruction is to find a set of shifts (*i,j,k*) (one for each image) that maximizes the global metric:

$$G(i,j,k) = \sum_{x=0}^{N} \sum_{y=0}^{M} \sum_{\substack{(x',y') \in \{(x+1,y),(x,y+1),(x+1,y+1)\} \\ x' \in [0,N] \lambda y' \in [0,M]}} C_{x,y,x',y'}(i_{x,y}, j_{x,y}, k_{x,y})$$

To solve this maximization problem, we used the optimization technique proposed in (*Griffiths et al., 1999*), which allowed finding the appropriate shifts with high accuracy (*Figure 2—figure supplement 1B*). All input 3D images were shifted according to the optimization results and registered using the multi-band blending approach (*Burt and Adelson, 1983*; *Brown and Lowe, 2003*).

## Bayesian algorithm for the detection of the surface of tissue sections

Most publicly available 3D image stitching methods were developed for EM data, where the samples are first embedded in resin or deep-frozen, which makes them solid and prevents partial removal of tissue by cutting. Therefore, they are based on local correlation of the images (*Saalfeld et al., 2012*; *Hayworth et al., 2015*). In the case of soft tissues, the removal of tissue upon cutting is much more significant, leading to a lack of texture correlations between two adjacent sections. The sample preparation process introduces several mechanical artefacts to the imaged sample, including uneven thickness of the section and tissue bending. When large vessels are aligned along the section surface, it becomes difficult to determine whether the empty space corresponds to the interior of the vessel or section damages or bending, which constitutes a major obstacle in their segmentation (*Figure 2—figure supplement 1C*). To address this issue, we propose a surface detection method, which uses prior distributions of expected section shape to find the border between the volume of the image of the sample (including blood vessels) and the out-of-field region.

Our approach is based on Bayesian statistics. According to the Bayes theorem:

$$p(y_1, y_2 | y_{m1}, y_{m2}) = \frac{p(y_{m1}, y_{m2} | y_1, y_2) p(y_1, y_2)}{p(y_{m1}, y_{m2})}$$

Using the chain rule to obtain the joint probability distribution $p(y_1, y_2)$, we got:

$$p(y_1, y_2 | y_{m1}, y_{m2}) \approx p(y_{m1}, y_{m2} | y_1, y_2) p(y_1 | y_2) p(y_1)$$

The empirical analysis of several tissue sections with manually specified surfaces allowed us to estimate the probabilities $(y_{m1}, y_{m2} | y_1, y_2)$, $p(y_1 | y_2)$ and $p(y_1)$ :

$$p(y_{m1}, y_{m2} | y_1, y_2) = \prod_{x,y} \frac{1}{\pi s \left(1 + \left(\frac{y_{1,x,y} - y_{m1,x,y}}{s}\right)^2\right)} \frac{1}{\pi s \left(1 + \left(\frac{y_{2,x,y} - y_{m2,x,y}}{s}\right)^2\right)}$$

$$p(y_1 | y_2) = \prod_{x,y} \frac{1}{\sqrt{2\pi}\sigma} exp\left(\frac{(y_{2,x,y} - y_{1,x,y})^2}{2\sigma^2}\right)$$

$$p(y_1) = \prod_{x,y} \prod_{\varepsilon x \in [-1,1]} \prod_{\varepsilon y \in [-1,1]} \lambda exp(-\lambda |y_{1,x+\varepsilon x, y+\varepsilon x} - y_{1,x,y}|)$$

Where $s$ is a parameter that specifies how close the real surface is to the measured one, $\sigma$ describes the variability of the section thickness, $\lambda$ specifies the smoothness of the real surface and $(x,y)$ are the coordinates of the real surface nodes.

By analysing our benchmark data set, we found that the median absolute deviation ($t_{MAD}$) of the section thickness $|y_{m2} - y_{m1}|$ constituted a good approximation for the parameters $s$ and $\sigma$. The parameter $\lambda$ was found by the maximum likelihood estimation of the empirical distribution measured from the maximum entropy segmentation. Then, the final posterior probability for surface detection has the following form:

$$p(ym_1, ym_2|y_1, y_2)$$

$$\approx \prod_{x,y} \frac{1}{\frac{\pi}{2}t_{MAD}\left(1 + \left(\frac{y_{1,x,y} - ym_{1,x,y}}{\frac{\pi}{2}t_{MAD}}\right)^2\right) \frac{\pi}{2}t_{MAD}\left(1 + \left(\frac{y_{2,x,y} - ym_{2,x,y}}{\frac{\pi}{2}t_{MAD}}\right)^2\right)}$$

$$\times \prod_{x,y} \frac{1}{\sqrt{2\pi}\frac{\pi}{2}t_{MAD}} exp\left(\frac{(y_{2,x,y} - y_{1,x,y})^2}{2\left(\frac{\pi}{2}t_{MAD}\right)^2}\right)$$

$$\times \prod_{x,y} \prod_{\varepsilon x\in[-1,1]} \prod_{\varepsilon y\in[-1,1]} \lambda_{ML}exp(-\lambda_{ML}|y_{1,x+\varepsilon x, y+\varepsilon x} - y_{1,x,y}|)$$

To check whether the surface energy model of this equation can be applied to different images, we created a benchmark data set composed of 10 section images with manually segmented surfaces. The model distributions $p(y_{m1}, y_{m2}|y_1, y_2)$, $p(y_1|y_2)$ and $p(y_1)$ closely matched with the corresponding empirical distributions calculated from the manual detection (*Figure 2—figure supplement 1D–F*).

The proposed model was used for the automated surface detection by minimizing the posterior $p(y_1, y_2|y_{m1}, y_{m2})$ probability . This minimization was performed using iterative conditional modes. The surfaces calculated by the maximum entropy approach were used as initial guess. To evaluate the quality of the automatically detected surfaces, we created a benchmark data set composed of 30 sections collected from three tissue samples. The average displacement between the manual and the automatic segmentation was 4.53 ± 1.12 voxels (*Figure 2—figure supplement 1G,H*).

## Segmentation of tissue-level networks

The goal of the segmentation of tissue-level networks is to identify the volume of a sample, which is occupied by large vessels such as CV, PV, hepatic artery or bile ducts. These structures appear in the images as empty volume (*Figure 2—figure supplement 2A*); therefore, their segmentation is possible without using a specific staining.

The direct application of thresholding methods like maximum entropy (*Kapur et al., 1985*) is troublesome due to several obstacles that arise from sample preparation artefacts. First, mechanical distortions such as uneven cutting of the section or tissue bending during imaging are introduced in the imaged sample. Since large vessels are not stained, it is impossible to distinguish them from out-of-field region using only the voxels intensities. Second, image intensities vary spatially within the sample due to uneven staining. In consequence, a global threshold underestimates the size of vessels in the bright regions of the image and overestimates it in the dark ones. To address these problems, we introduced two pre-processing steps. At first, we used the detected surfaces of the section to discriminate the parts of the image belonging to the sample from the ones in the out-of-field region. Subsequently, the 3D images (excluding the out-of-field region) were split into regular grids of overlapping sub-regions and the maximum entropy threshold was calculated for each of them. After that, the threshold values were interpolated over the entire image using tri-linear interpolation (*Figure 2—figure supplement 2B*). Finally, the vessels were segmented using the calculated threshold values (*Figure 2—figure supplement 2C*).

## Multi-resolution image positioning

Multi-resolution image positioning involves the rigid-body registration of a high-resolution 3D image (moving image) within the reconstructed low-resolution image of a section (fixed image). Since individual images have sizes up to 500 Mpx, we performed the image registration in the scaled-space using a stepwise approach.

We built a three-level scale pyramid using the original images and their copies scaled by factors of 0.50 and 0.25. The last level of the pyramid was used to find an initial approximation for the rigid-body registration, which was performed by rotating the moving image with respect to the fixed image. The rotation (r) with the highest value of cross-correlation was used as initial guess for further alignment.

Then, a registration based on polar transformations (*Wolberg and Zokai, 2000*) was performed. First, the relative shift between two images was found by the peak of their NCC, the images were shifted accordingly and its overlapping part was cropped. Second, the cropped images were transformed to polar coordinates (where a shift is equivalent to a rotation in the Cartesian coordinate system) and their NCC was calculated. The updated angle r was extracted from the peak of the cross-correlation of the transformed image. Note that the initial estimation of r ($\pm 15°$) found in the initial step is required for the convergence of the polar registration. The polar registration procedure was repeated subsequently using the images stored in the second and first level of scale pyramid, which results in the increase of the registration accuracy and computational time in each iteration of the algorithm. Also, 2–3 iterations were sufficient to achieve full convergence and register images with subcellular accuracy (*Figure 2—figure supplement 2D–F*).

## Methods for 3D image segmentation
### Nuclei splitting algorithm
In order to split artificially clustered structures, either the volumetric data from the segmented image or the triangle meshes of the reconstructed objects can be used (*Bilgin et al., 2013*). We used the information of the triangle meshes in a probabilistic algorithm, which first learns from the error distribution for the nuclei approximation by single and double ellipsoids. Based on the extracted statistics, the algorithm identifies and splits multi-nuclear structures. Further, we will refer to both the mono-, bi- and multi-nucleated structures as 3D objects.

First, all 3D objects were approximated by single and double overlapping ellipsoids. The first model corresponds to the minimum volume ellipsoid (MVE) that encloses the vertexes of the triangle mesh (*Figure 3—figure supplement 1D*). For the second model, the triangle mesh was symmetrically split in two subsets and each subset was approximated by an MVE (*Figure 3—figure supplement 1E*). Both models were evaluated on the data (vertexes) by using mean square error (MSE):

$$\text{MSE} = \frac{1}{n-9}\sum_{i=1}^{n}\left(\left(p_i - c\right)^T E(p_i - c) - 1\right)^2$$

where $n$ is the number of vertexes, $p_i$ is the coordinates vector of the vertex $i$, $c$ is the centre of the ellipsoid and $E$ is the matrix describing the orientation and dimensions of the ellipsoid. The model with the lowest MSE was selected as the best model for the 3D object.

Second, the error distribution (from the best models) resulting from the first step was analysed as follows: the natural logarithm of each MSE value was computed and the resulting histogram was fitted with a sum of two Gaussian distributions (*Figure 3—figure supplement 1F*). The two distributions were split by a threshold value, which was chosen such that it corresponded to the upper limit of the 95% confidence interval of the first component (the one with lowest mean value) (*Figure 3—figure supplement 1G*). The objects whose $ln(MSE)$ is smaller than the threshold corresponds either to one nucleus or two overlapping nuclei, and the rest corresponds to multi-nuclear structures. The 3D objects recognized as two overlapping ellipsoids were reconstructed using the models as boundaries to split the initially segmented images.

The multi-nuclear objects were split following two steps: first, the nuclei seeds were detected as proposed in (*Stegmaier et al., 2014*) and then the real shape of the nuclei was found by an active mesh expansion from the seeds. The nuclei seeds were extracted from the Laplacian of Gaussian scale-space maximum intensity projection (LoGMP) image (*Stegmaier et al., 2014*):

$$\text{LoGMP}(x, \sigma_{\min}, \sigma_{\max}) = \max_{\sigma_{\min} \leq \sigma \leq \sigma_{\max}} \text{LoG}(x, \sigma)$$

where LoG(x,$\sigma$) represents the Laplacian of Gaussian filtered image found using a standard deviation $\sigma$. Considering that the radius (r) of the objects to be detected is given by r=$\sqrt{2}\sigma$ (*Al-Kofahi et al., 2010*), $\sigma_{\min}$ and $\sigma_{\max}$ are determined by a priori knowledge of the typical size of the nuclei we want to detect. Each local maximum in the LoGMP image corresponds to a nuclei seed. Then, we used an active mesh expansion from the seeds to the real shape of the nuclei. The expansion was either limited to the nuclei border (regions of maximum intensity at the complement image of the LoGMP) or to the contact with neighbouring nuclei (*Figure 3—figure supplement 1H –J*).

## Methods for cell classification

### Feature extraction

For each nucleus, a profile of 74 parameters was calculated (*Table 1*). We used the information of the triangle mesh of the reconstructed nucleus as well as the information of the DAPI, Flk1 and phalloidin channels. All channel intensities were normalized using histogram equalization before the parameter extraction. The parameters include:

- Nuclear geometrical properties: volume ($V$), surface area ($A$), all possible ratios between the lengths of the semi-principal axis ($a$,$b$,$c$) of the MVE, sphericity ($\varepsilon = \frac{\pi^{1/3}(6V)^{2/3}}{A}$), mean and variance values of nucleus radius, shape index (*Levitt et al., 2004*) and curvature variation measure (*Sukumar et al., 2005*).
- DAPI and Flk1 intensity-based features:mean, standard deviation, skewness and kurtosis values of the intensity inside the nucleus.
- Haralick texture features (*Haralick et al., 1973*): The intensity of DAPI inside the nucleus was used. Thirteen statistical features were extracted from the normalized grey-level co-occurrence matrix, which was calculated from 65 independent co-occurrence matrices (considering all possible 13 directions in 3D and 5 different distances from 1 to 5 pixels). All co-occurrence matrices were calculated using 256 grey levels.
- Box-counting (BC) (*Lopes and Betrouni, 2009*) and Minkowski–Bouligand (MB) (*Einstein et al., 1998*) fractal dimensions: in both cases, the intensity of DAPI inside the nucleus was used. Integer values from 1 to 5 pixels were used as box length and radius values for the respective calculations. For theBC, the gliding box method was applied.
- Mean-weighted lacunarity (*Einstein et al., 1998*): it was calculated for the intensity of DAPI inside the nucleus using box lengths from 1 to 5 (lacunarity1, lacunarity2, etc.). Additionally, the values of the normalized lacunarity (NormLac2 = lacunarity2/ lacunarity1, etc.), and the natural logarithms of lacunarity (LogLac1, LogLac2, etc.), and normalized lacunarity (LogNormLac2, etc.) were extracted.
- DAPI mean surface intensity gradient: The surface gradient of DAPI signal was calculated at the centre of each triangle of the mesh. The mean value was calculated using a weighted average (using the area of the triangles as weights).
- Phalloidin and Flk1 intensity at different distances of the nucleus surface: the mean signal intensity (phalloidin or Flk1) at different distance (0 to 10 voxels) from the triangle mesh was calculated.

### Feature selection for the LDA

In order to get the most relevant parameter for the LDA classifier, we used the Fisher score and one-leave-out cross-validation as measure of the classifier accuracy. Firstly, the Fisher scores ($F_i$) was calculated for each parameter $i$ as follows:

$$F_i = \frac{\sum_{k=1}^{m} n_k (u_k^i - u^i)^2}{\sum_{k=1}^{m} n_k (\sigma_k^i)^2}$$

where $m$ is the number of classes, $n_k$ is the size of the kth class $u_k^i$ and $\sigma_k^i$ are the mean values and the standard deviation of the parameter $i$ for the kth class $u^i$ is the mean value of the parameter $i$ over the whole sample.

The parameters were sorted based on $F_i$ (*Table 1*) and systematically added to the classification while the accuracy of the algorithm was calculated, i.e., the first parameter from the sorted vector was taken, the classification was performed and the accuracy was calculated, then the second parameter was added and the process was repeated. *Figure 3—figure supplement 2B* shows how the classifier accuracy depends on the number of parameters used in the classification. For further analysis, only the set of parameters that yielded the highest accuracy was used.

The LDA was performed in three independent steps. Each corresponds to a two-class classification. First, hepatocytes were classified from other nuclei, then SECs were classified from the remaining nuclei and, finally, the rest of the nuclei were classified either into Kupffer or stellate cells.

## Cell classification by Bayesian network

The training set was presented as a vector of 75 parameters. The first one corresponded to the cell type and the following 74 were the measured nucleus features. Each parameter was discretized into 5 bins with equal population. Then, we calculated the mutual information *MI* between every parameter and the cell type parameter as

$$MI(X,Y) = \sum_{x,y} P(x,y) ln \left( \frac{P(x,y)}{P(x)P(y)} \right)$$

where *X* and *Y* denote sets of parameters, *x* and *y* denote instances of parameters. The probabilities were calculated from the training set as

$$P(x) = \frac{n_x + 1}{\sum_x n_x + r}$$

where $n_x$ denotes the number of instances in the bin *x* and *r* is the number of bins (in our case *r* = 5 for all parameters but the first one). Then the parameters were sorted in descent order according to the mutual information. The structure of Bayesian network was learned from the training data by the K2 algorithm (*Heckerman et al., 1995*) (*Figure 3—figure supplement 2C*). For each nucleus, the probability for each cell type was calculated. The type with the highest probability was taken as classification output.

## Validation of the resulting model

### Benchmark for the evaluation of 3D reconstructions of dense tissue

We generated a set of artificial images of liver tissue that can be used for developing and evaluating methods for the reconstruction of geometrical models of dense tissue. The benchmark consists of a set of realistic 3D high-resolution images (0.3 µm × 0.3 µm × 0.3 µm per voxel) of normal liver tissue. To generate artificial images that emulate the complexity of the real tissue images as well as exhibit meaningful biological characteristics, we extracted data from real images to produce idealized ground truth images of the main structures forming the tissue, that is, nuclei, sinusoids, BC and cell borders. Then, distortions coming from different sources such as uneven staining, optical distortion due to the PSF of the confocal microscope and spatially variation of Poisson noise were added to the idealized ground truth images (*Figure 3—figure supplement 5*).

The ground truth images were generated as follows: (1) the initial outlines were extracted from three real 3D images: central lines of the tubular networks (e.g. BC and sinusoids), position of the nuclei centres and cell borders; (2) idealized structures were built on top of the outlines: solid tubes with a constant radius of ~0.5 µm for the BC networks, hollow tubes with constant internal (~2.5 µm) and external (~3.0 µm) radius for the sinusoidal networks, solid spheres with radius between ~3.5 and ~5.5 µm for the nuclei and solid border of ~0.5 µm width for the cells.

Next, the uneven staining was simulated by applying random intensities at different scale levels. Briefly, first a 6 × 6 × 6 binning was applied to a black image and an intensity value extracted from a log-normal distribution with mean 1000 a.u. and standard deviation 0.5 was assigned to each binned voxel. Next, the image was unbinned and the new intensity values of each pixel were extracted from a log-normal distribution with mean equal to the original value and standard deviation 0.2. Finally, the uneven stained image was obtained by applying a mask (ground truth) to the generated one. Additionally, a homogeneous background (10:1, 4:1 and 2:1 signal-to-noise ratios) was added to the images (*Figure 3—figure supplement 5A,B*).

In thick samples far from the coverslip, the acquired images are highly distorted by the illumination PSF, leading to an asymmetric smearing of the image in z-direction (*Nasse et al., 2007*; *Nasse and Woehl, 2010*). We convolved the uneven stained images with realist PSFs (*Figure 3—figure supplement 5C,D*). The PSFs were generated using different excitations wavelengths for each structure: 568, 647, 780 and 488 nm for BC, sinusoids, nuclei and cell borders, respectively. Finally, we added Poisson noise with different scaling factors according to the models extracted from the real data (*Figure 3—figure supplement 5E–G*). An example of the resulting images is shown in *Figure 3—figure supplement 6*.

**Table 2.** Internal consistency of the sinusoidal network data.

| Sample | $V_s$ | $L_s$/(mm/mm$^3$) | $r_s$/(mm × 10$^3$) | $V_c \sim \pi \times r_s{}^2 \times L_s$ | $V_c/V_s$ |
|---|---|---|---|---|---|
| 1 | 0.16 | 2853.4 | 4.05 | 0.15 | 0.92 |
| 2 | 0.14 | 2976.4 | 3.75 | 0.13 | 0.95 |
| 3 | 0.20 | 3505.8 | 4.50 | 0.22 | 1.09 |
| *Hammad et al., 2014* | 0.15 | 5400.0 | 4.80 | 0.39 | 2.55 |

Notes: The fraction of volume of the sinusoids ($V_s$), the length of the sinusoidal network per volume unit ($L_s$) and the average radius of the network ($r_s$) were measured independently for each sample. A theoretical approximation of the fraction of volume of the sinusoids ($V_c$), considering it consists of ideal cylinders, was calculated ($V_c \sim \pi \times r_s^2 \times L_s$). Then, the ratio between the measured and the calculated fractions of volume ($V_c/V_s$) was calculated. Values close to 1.0 reflect auto consistency on the data. We performed the same calculation with the data reported in **Hammad et al. (2014)**.

## Internal consistency of the data extracted from sinusoidal network reconstruction

In order to check the internal consistency of the morphometric features that we extracted from the reconstructed sinusoidal networks, we independently calculated the fraction of volume of the sinusoids ($V_S$), the length of the sinusoidal network per volume unit ($L_S$) and the average radius of the network ($r_s$). Then, we estimated the fraction of volume of sinusoids ($V_C$) using $L_S$ and $r_s$, and approximating the tubular network by a cylindrical network. We found that $V_c/V_s = 0.99 \pm 0.09$, showing the internal consistency of our data. When applying the same calculation to the data reported in (*Hammad et al., 2014*), we found $V_c/V_s = 2.55$, which suggests an over-estimation of the network parameters (e.g. number of intersection nodes per mm$^3$, network length per mm$^3$); see *Table 2*.

## Quantitative analysis of liver tissue architecture

### DAPI integral intensity calculation

For each nucleus, the total content of DNA was calculated as the integral intensity of the original DAPI image inside the corresponding 3D triangle mesh. Since calculation was performed for three independent samples, the integral intensity per nucleus was normalized to the intensity of first one. Briefly, the distribution of DAPI integral intensity per nucleus was independently calculated for each sample (*Figure 4—figure supplement 2*). Then, each distribution was aligned (stretched) to the reference one (the first one in our case) by minimizing the functional:

$$d_j = \frac{\sum_i (f^0(x_i^0))^2 - \sum_i (f^0(x_i^0) \cdot f^j(s \cdot x_i^1))}{\sum_i (f^j(s * x_i^1))^2}$$

where $\sum_i$ is the sum over the bins of the distributions, $f^0(x_i^0)$ is the height of the i bin of the reference curve, $f^j(x_i^j)$ is the height of the i bin of the *j* curve to be aligned and s is the scaling (stretching) factor.

We found scaling factors 1.19 and 0.93 for the second and third samples respectively. Finally, the DAPI integral intensity of each nucleus was recalculated using the corresponding scaling factor.

## Acknowledgements

The authors acknowledge I. Sbalzarini, P. Tomancak and F. Jug (MPI-CBG) for comments on the manuscript. They also thank W. John and A. Muench-Wuttke from the Biomedical Services Facility for mouse care. Thanks also to J. Peychl for the management of the Light Microscopy Facility. This work was financially supported by the Virtual Liver initiative (http://www.virtual-liver.de), funded by the German Federal Ministry of Research and Education (BMBF), the Max Planck Society (MPG) and the DFG.

## Additional information

### Funding

| Funder | Author |
|---|---|
| Bundesministerium für Bildung und Forschung | Piotr Klukowski<br>Kirstin Meyer<br>Hidenori Nonaka<br>Giovanni Marsico |
| Max-Planck-Gesellschaft | Mikhail Chernykh<br>Alexander Kalaidzidis<br>Marino Zerial<br>Yannis Kalaidzidis |
| Deutsche Forschungsgemeinschaft | Kirstin Meyer |

The funders had no role in study design, data collection and interpretation, or the decision to submit the work for publication.

### Author contributions

HM-N, Developed the set of algorithms and statistics for cell classification, Developed the set of algorithms to create synthetic images, developed the network analysis statistics, Implemented the new and existing algorithms into a single platform, Conception and design, Analysis and interpretation of data, Drafting or revising the article; FS-M, KM, HN, MZ, Acquisition of data, Analysis and interpretation of data; PK, Developed the algorithm for the section surface detection, Analysis and interpretation of data, Drafting or revising the article; GM, MC, Implemented the new and existing algorithms into a single platform, Analysis and interpretation of data; AK, Transferred CPU implementation of the algorithms to GPU, Analysis and interpretation of data; YK, Conceived and directed the project, developed the de-noising algorithm, supervise the implemented the new and existing algorithms into a single platform, Conception and design, Acquisition of data, Drafting or revising the article

### Author ORCIDs

Hernán Morales-Navarrete, http://orcid.org/0000-0002-9578-2556

### Ethics

Animal experimentation: All animal studies were conducted in accordance with German animal welfare legislation and in strict pathogen-free conditions in the animal facility of the Max Planck Institute of Molecular Cell Biology and Genetics, Dresden, Germany. Protocols were approved by the Institutional Animal Welfare Officer (Tierschutzbeauftragter) and all necessary licenses were obtained from the regional Ethical Commission for Animal Experimentation of Dresden, Germany (Tierversuchskommission, Landesdirektion Dresden)(License number: AZ 24-9168.24-9/2012-1, AZ 24-9168.11-9/2012-3).

## Additional files

### Supplementary files

• Supplementary file 1. 3D 'ground truth' voxelated model of liver tissue (0.3 µm × 0.3 µm × 0.3 µm per voxel).

### Major datasets

The following datasets were generated:

| Author(s) | Year | Dataset title | Dataset URL | Database, license, and accessibility information |
|---|---|---|---|---|
| Morales-Navarrete H, Segovia-Miranda F, Klukowski P, Meyer K, Nonaka H, Marsico G, Chernykh M, Kalaidzidis A, Zerial M, Kalaidzidis Y | 2016 | A versatile pipeline for the multiscale digital reconstruction and quantitative analysis of 3D tissue architecture | http://dx.doi.org/10.5061/dryad.m67r6 | Available at Dryad Digital Repository under a CC0 Public Domain Dedication |

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
