## [Decision Letter]

Thank you for submitting your work entitled "A versatile pipeline for the multi-scale digital reconstruction and quantitative analysis of 3D tissue architecture" for consideration by *eLife*. Your article has been reviewed by two peer reviewers, and the evaluation has been overseen by Fiona Watt as the Senior Editor. One of the two reviewers has agreed to reveal his identity: Gaudenz Danuser.

The reviewers have discussed the reviews with one another and the editor has drafted this decision to help you prepare a revised submission.

The reviewers were very positive about your manuscript and would like to see it published in *eLife*. While the complexity of the synthetic data is appreciably lower than the experimental data, these simulations challenge the image analysis pipeline in a relevant way. Using these simulations as a test set for such algorithms is going to be helpful. We would even suggest that you make the code and parameter files available that generate these volumes for other labs to generate systematic other test data sets.

One concern we have is that the manuscript should be re-written to make it easier to follow. The main text should describe in broad strokes the necessary image analysis steps to a fairly naive audience. Currently, there is a large fluctuation in the level of detail – sometimes, unnecessary information is highlighted in the main text, whereas other, important, concepts are left out. The authors should then discuss the strengths and source of failures of the chosen approaches. For a resource article the failures are as interesting as the successes.

The authors should also avoid claims of exclusivity as much as possible. Individually, none of the image processing steps is out of the ordinary. The authors' achievement is to combine all the steps together into a working system. That said, the authors have failed to cite recent papers that address some of the same aspects. For example, it would be interesting to compare the recent work on nucleus segmentation with work published by Philip Keller's lab or Gaudenz Danuser's lab, and discuss Sean Megason's work on cell segmentation in 3D.

A second area of concern is the comparison of the new methods to other liver tissue reconstructions. The authors claim that substantial differences between the present and cited work originate in the advances of the image analysis algorithms. However, is this really the case? Could this not be due to differences in the tissue fixation or image acquisition? Some of the cited references are fairly recent. This might offer an excellent opportunity to apply the new image analysis pipeline to data that has been analyzed by other means. Such validation would be necessary to support the statement that 'in comparison with most recent previous reports, our model therefore provides a more precise estimation of the geometrical characteristics of sinusoidal and BC networks.'

---

## [Author Response]

The reviewers were very positive about your manuscript and would like to see it published in eLife. While the complexity of the synthetic data is appreciably lower than the experimental data, these simulations challenge the image analysis pipeline in a relevant way. Using these simulations as a test set for such algorithms is going to be helpful. We would even suggest that you make the code and parameter files available that generate these volumes for other labs to generate systematic other test data sets.

We thank the reviewers for this suggestion and included all necessary functions and scripts (scripts_image_generation.zip) for the generation of synthetic images as part of the MotionTracking (MT) software.

One concern we have is that the manuscript should be re-written to make it easier to follow. The main text should describe in broad strokes the necessary image analysis steps to a fairly naive audience. Currently, there is a large fluctuation in the level of detail – sometimes, unnecessary information is highlighted in the main text, whereas other, important, concepts are left out. The authors should then discuss the strengths and source of failures of the chosen approaches. For a resource article the failures are as interesting as the successes.

We modified the text and especially re-wrote the Results section to make it clearer and easier to follow. We added descriptions of general concepts as well as justifications for the use of the specific features of our pipeline. Specific technical details were moved to the Methods section.

The authors should also avoid claims of exclusivity as much as possible. Individually, none of the image processing steps is out of the ordinary. The authors' achievement is to combine all the steps together into a working system. That said, the authors have failed to cite recent papers that address some of the same aspects. For example, it would be interesting to compare the recent work on nucleus segmentation with work published by Philip Keller's lab or Gaudenz Danuser's lab, and discuss Sean Megason's work on cell segmentation in 3D.

We modified the text to avoid misunderstandings about claims of exclusivity. We now cite the published work, compare the results with ours and discuss recent work on nucleus and cell segmentation (Results, end of the subsection “3D image segmentation and active mesh tuning for the accurate reconstruction of tubular networks (sinusoids and BC) and nuclei”; Discussion, fourth paragraph; and Results, subsection “Cell classification and reconstruction”, last paragraph).

A second area of concern is the comparison of the new methods to other liver tissue reconstructions. The authors claim that substantial differences between the present and cited work originate in the advances of the image analysis algorithms. However, is this really the case? Could this not be due to differences in the tissue fixation or image acquisition? Some of the cited references are fairly recent. This might offer an excellent opportunity to apply the new image analysis pipeline to data that has been analyzed by other means. Such validation would be necessary to support the statement that 'in comparison with most recent previous reports, our model therefore provides a more precise estimation of the geometrical characteristics of sinusoidal and BC networks.'

We agree with the reviewers that the differences between our results and others on liver reconstruction could be due to not only differences in the image processing but also sample preparation and image acquisition. We moved this comparison from the Results section to the Discussion section (fifth paragraph) and expanded it. We applied our pipeline to a publicly available dataset (TiQuant: http://ms.izbi.uni-leipzig.de/index.php/software) (Hammad et al., 2014, Friebel et al., 2015). As shown in the Figure 7, our pipeline produces qualitatively more accurate segmentations than previous studies by others. In particular, our radius estimation for the sinusoidal network is closer to the values reported from EM data.

Author response image 1.Comparison of our pipeline with TiQuant (Hammad et al., 2014, Friebel et al., 2015) applied on the reconstruction of liver tissue.Our image analysis pipeline (MT) was applied on the dataset provided as a test for TiQuant ( http://ms.izbi.uni-leipzig.de/index.php/software). Panel (**A**) shows single 2D plane sections of a liver tissue image stained for sinusoids (magenta), whereas the segmentation generated by TiQuant (provided within the dataset) is shown as a partially transparent (green) image overlapping with the original one. Our reconstructions are shown as object contours (magenta lines). Panel (**B**) shows the radius distributions of the sinusoidal network extracted from reconstructions provided by TiQuant (red) and generated by the MT pipeline (blue). The estimated value of the radius is expressed in voxels since no information about the voxel size is provided. Each histogram was fit by a Gaussian distribution (with mean 14.08 and 11.00, for TiQuant and MT, respectively). The dots correspond to the extracted data and the solid lines to the corresponding fits. Our active mesh tuning approach yielded radius estimation for the sinusoidal network ~20% smaller than the voxel-based methods of TiQuant. These results are in line with the ones obtained from our experimental data.**DOI:**
http://dx.doi.org/10.7554/eLife.11214.035